# Aerosol organic nitrogen across the global marine boundary layer: distribution patterns and controlling factors

Ningning Sun[1,2], Xu Yu[2,3], Jian Zhen Yu[2,4*], Bo Zhang[1], Yilan Li[1,5], Ye Hu[1], Zhe Li[1], Zhenlou Chen[1], Guitao Shi[1*]

[1] Key Laboratory of Geographic Information Science (Ministry of Education), School of Geographic Sciences and State Key Laboratory of Estuarine and Coastal Research, East China Normal University, Shanghai, China

[2] Division of Environment and Sustainability, Hong Kong University of Science & Technology, Clear Water Bay, Kowloon, Hong Kong, China

[3] Jiangsu Key Laboratory of Atmospheric Environment Monitoring and Pollution Control, Jiangsu Collaborative Innovation Center of Atmospheric Environment and Equipment Technology, Joint International Research Laboratory of Climate and Environment Change, School of Environmental Science and Engineering, Nanjing University of Information Science and Technology, Nanjing, Jiangsu, China

[4] Department of Chemistry, Hong Kong University of Science & Technology, Clear Water Bay, Kowloon, Hong Kong, China

[5] Department of Chemistry, State University of New York College of Environmental Science and Forestry, Syracuse, NY, USA

*Correspondence:

Jian Zhen Yu (jian.yu@ust.hk)

Guitao Shi (gtshi@geo.ecnu.edu.cn)

**Abstract**

Organic nitrogen (ON) is an important yet poorly constrained component of aerosol total nitrogen (TN), particularly over remote oceans. We quantified aerosol ON in 92 total suspended particulate samples collected across approximately 160° of latitude in the marine atmospheric boundary layer (MABL) during Chinese Antarctic and Arctic expeditions (2019–2024), using a newly developed method that simultaneously determines ON and inorganic nitrogen. A significant latitudinal gradient was observed, with significantly higher ON concentrations (expressed as N) in the Northern Hemisphere ($83.3\pm141.4$ ng m$^{-3}$) than in the Southern Hemisphere ($15.4\pm12.4$ ng m$^{-3}$). Regionally, coastal East Asia recorded the highest ON levels ($164.6\pm179.1$ ng m$^{-3}$) but a lower ON/TN ratio ($21.1\pm7.9\%$), indicating strong terrestrial and anthropogenic influence. In contrast, the Arctic Ocean had lower ON concentrations ($19.1\pm19.0$ ng m$^{-3}$) but the highest ON/TN ratio ($38.6\pm12.4\%$), suggesting dominant marine biogenic sources. The Southern Ocean showed the lowest ON concentration ($12.0\pm7.1$ ng m$^{-3}$) yet a relatively high ON/TN ratio ($27.8\pm11.0\%$), also pointing to oceanic origins. Near Antarctica, samples influenced by sea-ice air masses displayed markedly elevated ON and ON/TN ratios. These increases were strongly correlated with sea ice concentration and chlorophyll-a exposure, indicating enhanced biogenic emissions from sea-ice-associated ecosystems. This study offers the first direct ON measurements along a global MABL transect, revealing distinct latitudinal and regional patterns, and emphasizing the combined roles of continental inputs and marine sources. It also identifies sea-ice dynamics as a key factor influencing ON in Antarctic regions, providing crucial data for improving atmospheric and climate models.

## 1. Introduction

Marine atmospheric boundary layer (MABL) aerosol particles contain significant amounts of organic nitrogen (ON) and inorganic nitrogen (IN), both recognized as major components of atmospheric particulate matter (Li et al., 2023). ON may account for roughly 20–80% of total reactive nitrogen deposition to the surface ocean, implying a potentially large, yet uncertain, role in marine nitrogen cycling and climate (Altieri et al., 2016, 2021). ON affects climate and biogeochemistry by supplying bioavailable nitrogen, modifying cloud condensation nuclei and ice-nucleating particle populations, and contributing to aerosol light absorption. Hygroscopic ON compounds (e.g., amino acids, amines, sugars) enhance water uptake and cloud condensation nuclei (CCN) activity; some proteinaceous organics act as efficient ice nuclei (Alsante et al., 2024; Chan et al., 2005). Marine alkylamines can form salts with sulfuric acid, promoting new particle formation and growth, thereby linking ON to aerosol number and radiative forcing (Almeida et al., 2013; Brean et al., 2021). Nitrogen-containing chromophores (brown nitrogen) can dominate the absorptive properties of organic aerosol regionally and contribute substantially to global absorption by carbonaceous aerosol (Li et al., 2025).

However, ON remains poorly constrained due to analytical limitations (Baker et al., 2017). Previous studies focused on the water-soluble fraction of aerosol ON (WSON) inferred indirectly by subtraction IN from total nitrogen (TN) (ON = TN – IN), while the water-insoluble organic nitrogen (WION) fraction has been largely unquantified (Cornell, 1999; Mace et al., 2003). The subtraction approach is prone to errors and artifacts, especially when TN and IN concentrations are similar, leading to underestimation and large uncertainties in ON burdens and fluxes. A novel method developed by Yu et al. (2021) addresses these limitations. Based on thermal evolution and chemiluminescence detection, this approach measures aerosol IN and ON simultaneously, eliminating subtraction-based biases and capturing both WSON and WION.

Aerosol ON arises from diverse sources. Marine pathways include primary

emissions via sea spray enriched with organic matter from the sea surface microlayer
and secondary formation from marine precursors (e.g., alkylamines) reacting with
acidic species (Facchini et al., 2008; Miyazaki et al., 2011a). Continental pathways
include long-range transport of organic emissions from fossil fuel combustion,
biomass burning, soils, and vegetation (Cape et al., 2011; Jickells et al., 2013; Luo et
al., 2018). Primary marine emissions inject large amounts of particulate matter
annually, carrying organic carbon and nitrogen from plankton, bacteria, and surface
films (Violaki et al., 2015a). Observations have shown that sea spray can carry
substantial ON and that WION can dominate ocean-influenced aerosol ON (Miyazaki
et al., 2011a).
While marine aerosol ON has been the subject of several studies, its sources in
remote oceanic regions remain a matter of debate. Some studies implicate continental
transport (e.g., dust, anthropogenic emissions), whereas others point to direct sea
spray emissions or secondary formation from marine-derived alkylamines (Altieri et
al., 2016; Lesworth et al., 2010; Zamora et al., 2011). Correlations between ON and
ocean biological proxies (e.g., chlorophyll-a) suggest in situ marine production,
particularly during phytoplankton blooms (Altieri et al., 2016; Dall'Osto et al., 2019).
Yet open-ocean and polar regions, where sea ice variability can strongly modulate
primary productivity and thus potentially influence ON emissions, remain sparsely
observed, limiting constraints on potential sea ice linked controls on ON, especially
for high latitudes (Altieri et al., 2016; Matsumoto et al., 2022). Around Antarctica in
particular, the paucity of direct ON measurements—especially of WION—limits
understanding of ON sources, seasonality, and impacts on high-latitude atmospheric
chemistry.
To address these gaps, we measured aerosol ON and IN using samples collected
during four Chinese Arctic and Antarctic research expedition campaigns, spanning
~160° of latitude from the Arctic to Antarctica. The dataset, determined by this newly
developed analyzer, enables evaluation of hemispheric and regional patterns,
assessment of controlling factors (e.g., continental influence, marine biological
activity), and explicit investigation of sea-ice–associated processes near Antarctica.

The results provide observational constraints that can be used to refine the representation of nitrogen cycling and atmosphere–ocean interactions in climate and atmospheric chemistry models.

## 2. Methodology

### 2.1. Sample Collection

A total of 92 total suspended particulate (TSP) samples were collected during three Chinese Antarctic research expeditions and one Arctic expedition aboard the icebreaker R/V *Xuelong*. Sampling spanned a latitudinal range of approximately 160° (86°N to 75°S), encompassing polar and mid-latitude marine regions. The Antarctic samplings were conducted in October to November in 2019 (SP2019, 14 samples), November 2021 to March 2022 (SP2021, 23 samples), and October 2023 to April 2024 (SP2023, 15 samples), while the Arctic campaign occurred in July to September in 2021 (40 samples).

During the Antarctic campaigns, aerosols were collected using a high-volume air sampler (HVAS, TISCH Environmental, USA; flow rate: 1.2 $m^3$ $min^{-1}$) equipped with pre-baked (500°C, 24 h) Whatman quartz filters (20.3 × 25.4 cm; Whatman Ltd., UK). For Arctic sampling, a DIGITEL DHA-80 sampler (flow rate: 500 L $min^{-1}$) with 14.2 cm diameter Whatman quartz filters were employed. Each sample represented a 48 h integrated collection period, corresponding to 2–4° latitude traversed during ship transits. To minimize contamination from ship emissions, a wind sector controller restricted sampling to air masses within 120° of the ship's heading. Filters were handled using nitrile gloves and masks to avoid potential contamination. Post-sampling, filters were folded with the collection surface inward, wrapped in pre-cleaned aluminum foil, sealed in polyethylene bags, labeled with sampling time and location, and stored at -20°C. Detailed protocols followed established methodologies (Shi et al., 2021). Following expeditions, samples were transported to the laboratory under frozen conditions and maintained at -20°C until analysis. The sampling location for the Antarctic and Arctic campaigns are illustrated in Fig. 1.

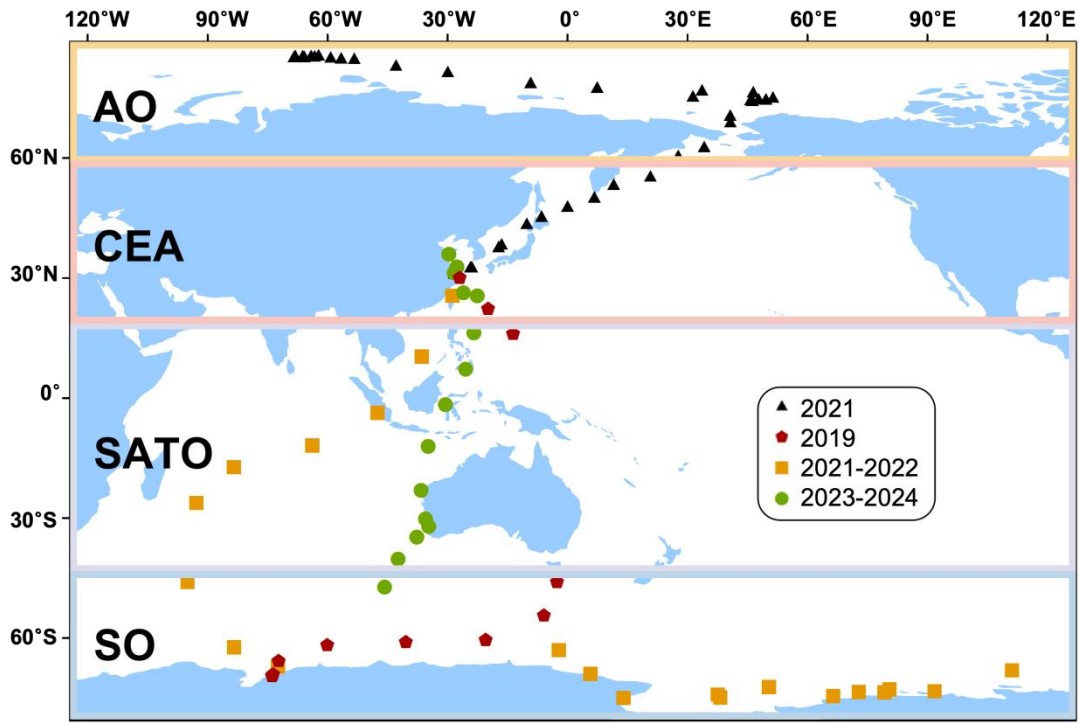

137

Figure 1. Total suspended particulate (TSP) aerosol sampling locations along the cruises path from Shanghai, China to Antarctica and Arctic.

## 2.2. Chemistry Analysis for major ions, EC and OC

Major ions were quantified through ion chromatographic analysis of water extracts of the aerosol samples. The extraction of filters in the laboratory followed protocols comparable to those described in the previous study (Shi et al., 2021). Prior to measurement, three-quarters of each filter was sectioned into small pieces using acid-cleaned Teflon-coated scissors and transferred into high-purity Milli-Q water (18.2 M$\Omega$). The suspensions were subjected to ultrasonic treatment for 30 min, followed by continuous shaking at 120 rpm for 12 h to ensure thorough extraction of water-soluble components. The extracts were subsequently filtered through 0.22 $\mu$m polytetrafluoroethylene (PTFE) membranes prior to ion analysis. The concentrations of the main ions ($NO_3^-$, $SO_4^{2-}$, $Na^+$, $NH_4^+$, $K^+$, and $Ca^{2+}$) in the sample were determined by an ion chromatograph (AQ1100, RFIC, equipped with a CS12 column (2×250 mm) for cation analysis, AS11 column (2×250 mm) for anion analysis, Thermo Scientific, USA), and the eluents of cation and anion were 18.00 mM methylsulfonic acid (MSA) and potassium hydroxide (KOH), respectively. During

sample analysis, the relative deviation of repeated assays (n = 5) of all ions is usually
less than 5%. We used the following formula to calculate non-sea salt $SO_4^{2-}$
($nssSO_4^{2-}$), non-sea salt $Ca^{2+}$ ($nssCa^{2+}$) and non-sea salt $K^+$ ($nssK^+$):

$$[nssSO_4^{2-}] = [total\ SO_4^{2-}] - 0.253 \times [Na^+] \qquad (1)$$

$$[nssCa^{2+}] = [total\ Ca^{2+}] - 0.038 \times [Na^+] \qquad (2)$$

$$[nssK^+] = [total\ K^+] - 0.037 \times [Na^+] \qquad (3)$$

where 0.252, 0.037, and 0.038 in the above expressions are the ratios of $SO_4^{2-}/Na^+$
(Quinby-Hunt and Turehian, 1983), $Ca^{2+}/Na^+$ (Anonymous, 1997), and $K^+/Na^+$ (Keene
et al., 1986) in the sea water, respectively.
OC and EC concentrations were determined using a Thermal/Optical Carbon
Analyzer (DRI, Model 2001, Atmoslytic Inc., USA) following the IMPROVE
protocol as implemented by Wu et al., (2024). OC and EC measurements were
conducted for aerosol filters collected during the 2021 Arctic and 2019 Antarctic
cruises.

**2.3. ON measurement**

Aerosol ON and IN were simultaneously measured using the recently developed
Aerosol Nitrogen Analyzer system, which enables sensitive quantification directly
from filter samples without pretreatment. Detailed descriptions of the method are
provided in Yu et al (2021). Briefly, the method detection limit is 96 ng N. Because
the detection limit scales inversely with the analyzed filter area, it can be readily
lowered by analyzing a larger aliquot. In this study, 4–6 cm$^2$ of filter material was
typically analyzed for each sample, yielding a proportionally lower effective detection
limit and ensuring stable and reliable quantification for low-concentration marine
aerosol samples. Compared with traditional IC-based approaches, this analyzer
provides a clear advantage by determining IN and ON simultaneously on the same
filter aliquot, thereby avoiding the subtraction-based "difference method" (ON = TN −
IN) and the associated uncertainty propagation when TN and IN are similar in
magnitude.
The analyzer integrates a thermal aerosol carbon analyzer and a
chemiluminescence $NO_x$ analyzer. Aerosol samples collected on quartz fiber filters
were thermally evolved under a programmed 6-step temperature protocol (150, 180,
300, 400, 500, and 800 °C) in a 1% $O_2$/99% He carrier gas. The evolved materials
were catalytically oxidized to $CO_2$ and nitrogen oxides ($NO_y$), with the C signal
monitored via methanator-FID detection and the N signal recorded through
chemiluminescence after converting $NO_y$ to NO. The C signal assists in
differentiating IN and ON components, as ON aerosols produce both C and N signals
while the IN fraction only yields an N signal. The programmed thermal evolution
facilitates separation of aerosol IN and ON due to their distinct thermal characteristics.
Specifically, IN and ON discrimination is achieved by jointly interpreting the C and N
thermograms: ON is identified by co-evolving C and N signals across the temperature
steps, whereas IN is characterized by N-only evolution without a corresponding C
signal. The separation of overlapping thermal features is further resolved using
multivariate curve resolution (MCR), which deconvolves the mixed thermograms into
source-like components based on their distinct thermal evolution patterns.
Quantification of IN and ON is achieved through multivariate curve resolution (MCR)
data treatment of the C and N thermograms using USEPA PMF (version 5.0).

## 198     2.4. Backward Trajectory Analysis

To study air mass origins, air mass backward trajectories have been calculated using
the Hybrid Single-Particle Lagrangian Integrated Trajectories (HYSPLIT) model with
meteorological fields from the National Oceanic and Atmospheric Administration
(NOAA) air resources laboratory GDAS database. Five-day backward trajectories
were calculated in order to reveal the history of the air masses arriving at the sampling
site (Stein et al., 2015). Each trajectory originated at the vessel's real-time position
with an arrival height of 20 m, capturing boundary layer transport while minimizing
local ship influence. Air mass backward trajectories were simulated using the
HYSPLIT model with meteorological fields from the NOAA GDAS database to
reveal the transport history of air masses arriving at the vessel (Stein et al., 2015).
Given that the ship was continuously moving and each sample integrates air masses

over approximately 2–4 degrees of latitude, we applied a nested strategy to account for spatiotemporal variability. For the initial characterization of the entire dataset, a representative sampling location was defined for each sample using the average latitude and longitude of its start and end positions, with backward trajectories simulated at 6 h intervals anchored to this midpoint to identify dominant air-mass categories (Fig. 3). Subsequently, to precisely investigate the influence of sea ice on ON in the Southern Ocean and Antarctic marginal regions (Section 4.2), a targeted high-resolution analysis was performed on this subset of samples. For each Antarctic sample, the actual cruise track was equally divided into 48 points corresponding to the hourly intervals of the 48 h sampling period, and a 120 h backward trajectory was calculated for each of these 48 coordinates (Fig. S3a and b).

To determine whether the backward trajectories of the MABL samples were mainly influenced by the open ocean, sea-ice-covered regions, or the continental area, we calculated the time-weighted residence-time ratios of air masses over sea ice ($R_S$), open ocean ($R_O$), and the continental area ($R_C$) using the following equation:

$$R_S(R_O \text{ or } R_C) = \frac{\sum_{i=1}^{N_S(N_O \text{ or } N_C)} \times e^{-(\frac{t_i}{120})}}{\sum_{i=1}^{N_{total}} \times e^{-(\frac{t_i}{120})}} \tag{4}$$

where $N_{total}$ denotes the total number of trajectory endpoints; $N_S$ $N_O$ and $N_C$ represent the numbers of endpoints located over sea ice, the open ocean, and the Antarctic ice sheet, respectively. $t_i$ is the backward-trajectory time (in hours), and $t_i/120$ is a time-weighting factor (Zhou et al., 2021). This factor accounts for air-mass dispersion during transport and aerosol removal by particle deposition; therefore, regions associated with longer trajectory times exert weaker influences on the sampling site, whereas nearby regions exert stronger influences. Accordingly, higher values of $R_S$, $R_O$ and $R_C$ indicate greater influences from sea ice, the open ocean, and the Antarctic ice sheet, respectively.

**2.5. Potential Source Contribution Function (PSCF) analysis**

Potential Source Contribution Function (PSCF) analysis was implemented to identify source regions of ON observed during the sampling period (Ashbaugh et al., 1985). A

higher PSCF value indicates a greater potential source contribution to the receptor site. In our study, the PSCF domain was established within a grid cell encompassing all backward trajectories. The cruises were discretized into 1° latitude × 1° longitude grid cells. The PSCF value for cell ij was calculated as:

$$PSCF_{ij} = \frac{\sum m_{ij}}{\sum n_{ij}} \quad (5)$$

where, $m_{ij}$ = total trajectory endpoints within cell ij; $n_{ij}$ = subset of endpoints associated with aerosol component concentrations exceeding the 75th percentile of cruise measurements. To mitigate uncertainty in cells with sparse trajectory density, a latitude-dependent weighting function (W) was applied:

$$W = \begin{cases} 1.0 \text{ when } n_{ij} > N2 \\ 0.8 \text{ when } N1 < n_{ij} < N2 \\ 0 \text{ when } n_{ij} < N1 \end{cases} \quad (6)$$

where $n_{ij}$ is the number of trajectories passing for each cell in the study period and $N1 = 60*\cos(latitude)$, and $N2 = 300*\cos(latitude)$. The cosine factor is used to account for the changing grid cell size with varying latitude.

## 2.6. Air-mass exposure to chlorophyll a

The Air-mass Exposure to Chlorophyll a (Chl-a) index (AEC) serves as a quantitative metric to assess the influence of marine biogenic emissions on a target region through air mass transport (Blazina et al., 2017; Choi et al., 2019). This approach is grounded in the well-established correlation between ocean surface phytoplankton biomass and marine biogenic emissions, particularly dimethyl sulfide (DMS), where Chl-a concentration acts as a robust proxy for phytoplankton abundance (Siegel et al., 2013). The AEC index estimates the integrated exposure of an air mass to oceanic DMS source regions along its trajectory by accounting for both spatial distribution of Chl-a and atmospheric vertical mixing dynamics (Zhou et al., 2023).

For each trajectory point, Chl-a concentrations (Chla$_i$) were obtained from satellite remote sensing products (Aqua-MODIS, OCI algorithm; 8-day composite, 4 km × 4 km resolution; https://oceancolor.gsfc.nasa.gov/l3/) within a 20 km radius to reduce the influence of missing/cloud-contaminated pixels and pixel-scale noise, while remaining small enough to preserve local marine biological variability relevant

to each trajectory point. The 20 km radius approach has been widely adopted in
previous studies to mitigate the uncertainty of trajectory endpoints and ensure robust
matching with satellite data coverage in previous research (Park et al., 2018; Zhou et
al., 2021, 2023). Trajectory endpoints over Antarctica, sea-ice-covered areas, or at
pressures < 850 hPa were assigned Chl-a = 0 because air masses at these altitudes are
generally decoupled from local ocean surface biological activity (Zhou et al., 2023).
Points without valid Chl-a data were excluded. The AEC for a single trajectory was
computed as:
$$AEC = \frac{\sum_{i=1}^{120} Chla_i \times e^{-(\frac{t_i}{120})}}{n} \quad (7)$$
where $t_i$ denotes time backward along the trajectory (hours), and n is the total number
of valid trajectory points. The time points when the air mass passed over the continent
or regions covered by sea ice were assigned a zero chlorophyll value. To ensure
robustness, trajectories with n < 90 (75% of 120 h data points at hourly resolution)
were discarded. For each sample, the final AEC value was derived from the arithmetic
mean of all valid trajectories during the sampling period (Yan et al., 2024).
**2.7. Sea ice concentration**
In this study, remote sensing data are utilized to illustrate the spatiotemporal
distribution of sea ice concentrations (SICs) in the Southern Ocean. For regional-scale
visualization of sea-ice extent (SIE) and SIC variability, we used the Sea Ice Index
(Version 3) distributed by the National Snow and Ice Data Center (NSIDC) (Fetterer
et al., 2017), which is derived from passive-microwave observations from DMSP
SSM/I and SSMIS sensors (Cavalieri et al., 1997).
Sea-ice concentrations used here are derived from daily gridded
passive-microwave SIC products, which provide all-weather coverage and are widely
used for polar sea-ice monitoring. The SIC of each sample is calculated using the
following formula:
$$SIC = \frac{\sum_{i=1}^{Ns} SIC_i}{Ns} \quad (8)$$
where $SIC_i$ represents the average sea ice density at the endpoint of the specified track.
Ns represents the total number of trajectory endpoints located on the sea ice area. For
each trajectory endpoint, the SIC value was extracted by collocating the endpoint
latitude/longitude and the corresponding day with the daily SIC grid; the $SIC_i$ for each
sample was then calculated as the mean SIC across all sea-ice-covered endpoints (Ns).
Sea ice concentration data are from the AMSR2 dataset (Version 5.4. University of
Bremen, Germany. Index of /amsr2/asi_daygrid_swath/s3125).

## 3. Results

Atmospheric ON concentrations (expressed as N, the same hereafter) exhibited
significant hemispheric differences ($p < 0.001$; Mann–Whitney U test; Table S1), with
values in the Northern Hemisphere (NH: $83.3 \pm 141.4$ ng m$^{-3}$, N = 55) being
approximately five times higher than those in the Southern Hemisphere (SH: $15.4 \pm$
$12.4$ ng m$^{-3}$, N = 37). The ON/TN ratios showed broadly similar magnitudes between
hemispheres, with slightly higher in the NH ($30.4 \pm 13.6\%$) compared to the SH ($27.9$
$\pm 10.6\%$). Samples from three Antarctic cruises—SP2019 (mean = 19.4 ng m$^{-3}$; range:
9.5–555.6 ng m$^{-3}$), SP2021 (mean = 20.4 ng m$^{-3}$; range: 1.3–81.3 ng m$^{-3}$), and
SP2023 (mean = 18.3 ng m$^{-3}$; range: 1.8–457.0 ng m$^{-3}$) showed no significant
variation (one-way ANOVA; $p > 0.2$), indicating that interannual variation was rather
minor. A clear latitudinal gradient in ON concentrations was observed along the
Antarctic-to-Arctic transect, with peak values in the 20–40° N zone and a gradual
decline toward both polar regions (Fig. 2a). Based on spatial distribution patterns, the
study transect can be divided into four regions (Fig. 1): (1) the Arctic Ocean region
(AO, north of ~60° N); (2) the Coastal East Asia region (CEA, 20–60° N); (3) the
Southeast Asia-Australia Tropical Ocean region (SATO, ~ 20° N–40° S); and (4) the
Southern Ocean region (SO, south of ~ 40° S).
The CEA region exhibited the highest ON concentrations (mean = 164.6 ng m$^{-3}$)
but the lowest ON/TN ratio (mean = $21.1 \pm 7.9\%$). In contrast, the SO region showed
the lowest ON concentrations (mean = 12.0 ng m$^{-3}$; range: 1.8–32.3 ng m$^{-3}$) and
higher ON/TN ratios (mean = $27.8 \pm 11.0\%$). Notably, the AO region displayed the
highest ON/TN ratios (mean = $38.6 \pm 12.4\%$) despite relatively low ON
concentrations (mean = 19.1 ng m$^{-3}$; range: 5.2–32.2 ng m$^{-3}$). The ON/TN ratio in
SATO region (26.8 ± 10.0%) is similar to that of SO but with a lower ON
concentration (mean = 23.4 ng m$^{-3}$; range: 5.7–70.1 ng m$^{-3}$), which is much lower
than the CEA region, but higher than the high latitude two pole regions.

326       Since direct measurement data of total ON in global MABL are limited, WSON

data were summarized for comparison (Table 1). Overall, the previous results are
consistent with the spatial trends of ON in our study. WSON concentrations exhibit
significant spatial variation, generally higher in the NH than in the SH, highlighting
the substantial contribution of anthropogenic sources (Violaki et al., 2015b). In
addition, WSON concentrations tend to be higher closer to land, while in remote
ocean areas, WSON levels are generally lower. The reported ratios of WSON/WSTN
in previous studies vary significantly across different investigation sites. Moreover, in
remote marine environments, the WSON/WSTN ratio is relatively high, suggesting
that WSON plays a substantial role in the biogeochemical cycle of nitrogen within
these remote regions. It is important to note that most previous studies over the
remote ocean measured only WSON, without accounting for the WION. As a result,
the ON/TN ratios in this region were likely underestimated. Based on our comparison,
the total ON concentration in the Southern Ocean may have been underestimated by
approximately 40%, hinting the significant contribution of the insoluble organic
fraction that has been largely overlooked in earlier datasets due to measurement
method limitations.

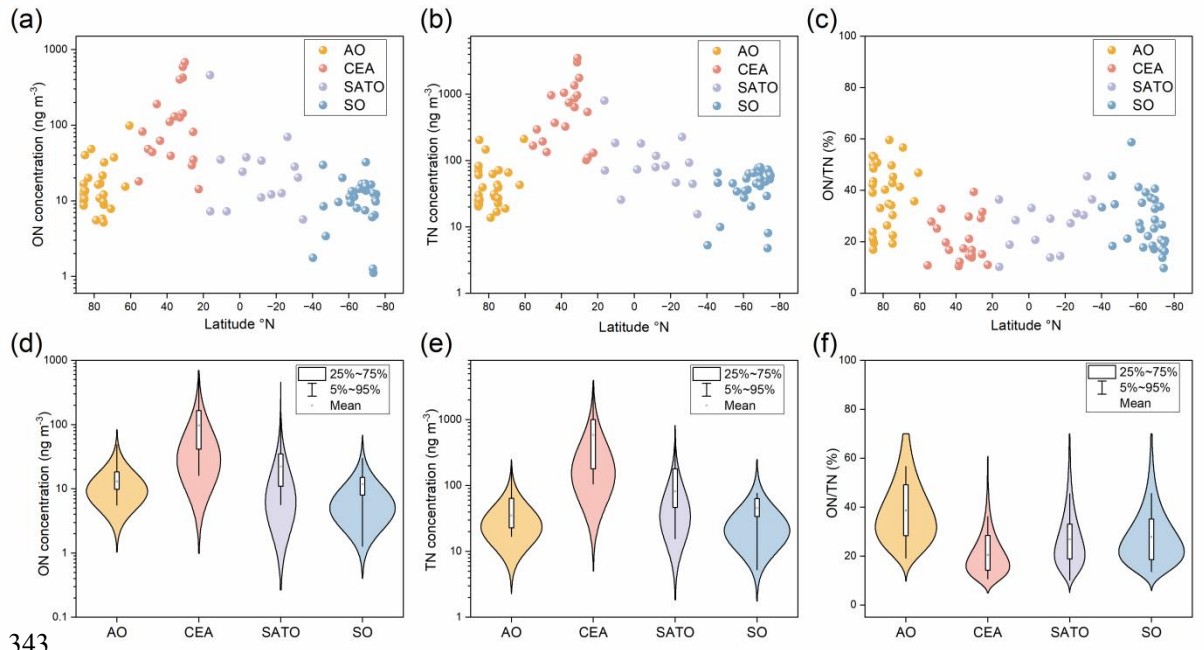


Figure 2. Latitudinal distributions of ON concentration, TN concentration and ON/TN ratio (a, b,
c), and the statistics (d, e, f) over the the Arctic Ocean region (AO), the Coastal East Asia region
(CEA), the Southeast Asia-Australia Tropical Ocean region (SATO) and the Southern Ocean
region (SO), respectively.

Table 1. Measured concentrations of WSON from published reports in the marine atmospheric boundary layer.

| Regions | Period | Locations | WSON (ng m$^{-3}$) | Ratio to WSTN(%) | Methods for IN | Methods for WSTN | Reference |
|---|---|---|---|---|---|---|---|
| Southern Atlantic | 2007.01-02 | 35° S–45° S | 119±163.8 | - | IC | PO | (Violaki et al., 2015b) |
| the Northwest Pacific Ocean | 2014.2015 | 25–40° N, 125–150° E | 43.4–564.2 | 11–46 | IC | PO | (Luo et al., 2018) |
| the coast of China the East China seas | 2014.2015 | 30–40° N, 120–130° E | 96.6–7238 | 6–48 | IC | PO | (Luo et al., 2018) |
| Southern Ocean | 2000.11 | 40.41° S, 144.41° E | 50.4±79.8 | 21 | IC | UV | (Mace et al, 2003) |
| Oahu, Hawaii | 1998.7.22-8.13 | 21.7°N, 157.8° W | 46.2 | 31 | IC | UV | (Cornell et al., 2001) |
| the southern margin of the East China Sea | 2005-2006 | 25.09° N, 121.46° E | 476±756 | 24±16 | IC | UV with PO | (Chen and Chen, 2010) |
| the western North Pacific | 2008.08.24-09.13 | 42.98° N,144.37° E −35.65° N, 139.77° E | 130±61 (10−260) | 67±15 | IC | TOC/TN analyzer | (Miyazaki et al., 2011b) |
| Huaniao Island | 2019 | 30.86° N, 122.67° E | 30–2810 | 0.13−77 | IC | TOC/TN analyzer | (Tian et al., 2023) |
| the northern tip of Japan | 2010-2012 | ~45.2° N, ~141.2° E | 77±57 | 12.8±15.2 | IC | TOC/TN analyzer | (Matsumoto et al., 2017) |
| the Southern Ocean | 2016-2020 | 38.8–69.0° S, 38.1–150.8° E | 4.7 | 20 | IC | TOC/TN analyzer | (Matsumoto et al., 2022) |
| the Subarctic Western North Pacific Ocean | 2016.7.21-8.22 | 30–65° N, 130–160° W | 1.62–205.8 | 9 | IC | TOC/TN analyzer | (Jung et al., 2019) |
| Bermuda | 2011 | 32.27° N, 64.87° W | 105±191.8 | 50.4±18.9 | nutrient analyzer | TN Analyzer | (Altieri et al., 2016) |
| the Arctic Ocean | 2021.07-09, 2021.07-09, | north of ~60° N | 5.2–32.2 | 38.6±12.4 | N Analyzer | N Analyzer | this study |
| the Coastal East Asia | 2019.10-11, 2023.10-2024.04 | 20–60° N | 18.1–555.6 | 21.1±7.9 | N Analyzer | N Analyzer | this study |
| the Southeast Asia-Australia Tropical Ocean | 2021.11, 2023.11 | 20° N–40° S | 5.7–70.1 | 26.8±10.0 | N Analyzer | N Analyzer | this study |
| the Southern Ocean | 2021.11-2022.03, 2019,11, 2023.11 | south of ~40° S | 1.8–32.3 | 27.8±11.0 | N Analyzer | N Analyzer | this study |

*PO: the persulfate oxidation (PO) method
*UV: ultraviolet photo-oxidation
*TN analyzer: a total organic carbon (TOC) analyzer with a TN unit
*Nutrient analyzer: automated nutrient analyzer and standard colorimetric method

## 4. Discussion

### 4.1 Source identification of ON

ON in the MABL primarily originates from two main source pathways: marine emissions and long-distance continental transport. Marine sources include primary ON, predominantly associated with sea-spray particles enriched in biological material from the ocean surface microlayer, and secondary ON. The latter not only derives from marine precursors such as alkylamines that react with acidic species (Altieri et al., 2016; Facchini et al., 2008), but also significantly involves the atmospheric oxidation of marine-derived biogenic volatile organic compounds (BVOCs). Specifically, isoprene and monoterpenes emitted from the ocean can react with hydroxyl (OH) or nitrate radicals ($NO_3$) to form secondary organic nitrates (Fisher et al., 2016; Ng et al., 2017). Additionally, direct sea-to-air emissions of light alkyl nitrates produced photochemically in the surface water contribute to the MABL ON pool (Chuck et al., 2002). Continental sources involve the long-range transport of organic emissions—including combustion byproducts, soil- and vegetation-derived compounds, and biomass burning aerosols. It is important to note that these continental inputs include both ON formed directly over land and ON produced from continental precursors during transport (Duce et al., 2008; Li et al., 2025). This transport can significantly influence remote ocean regions (Cape et al., 2011; Jickells et al., 2013).

ON concentrations in the CEA region were the highest among all study regions, with air masses spending 22.6% of their 5-day trajectories over continental areas (Fig. 3b). A significant correlation between ON and crustal elements such as $nssCa^{2+}$ (r = 0.75, $p$ < 0.01; Fig. 4) likely suggests the influences of continental transport of particles on the ON levels in this region (Xiao et al., 2016). A significant correlation between ON and the anthropogenic tracer EC (r = 0.81, $p$ < 0.01; Fig. 4) indicates that fossil fuel combustion and biomass burning are important ON sources (Shubhankar and Ambade, 2016; Wu and Yu, 2016). Similarly, the robust association between ON and $nssK^+$ (r = 0.78, $p$ < 0.01; Fig. 4), a tracer of biomass burning, also supports contributions from agricultural and residential biomass burning (Song et al., 2018). Despite the high absolute ON concentrations, the relatively low ON/TN ratio (21.1%) likely reflects disproportionately elevated IN emissions from intensive human activities, particularly $NH_3$ volatilization from agriculture and vehicular $NO_x$

emissions (Pavuluri et al., 2015). This interpretation aligns with emission inventories
that identify the CEA as a global nitrogen pollution hotspot, where ON is co-emitted
or formed from precursors that share common sources with EC and other
combustion-related pollutants, originating from incomplete combustion and industrial
processes (Deng et al., 2024).

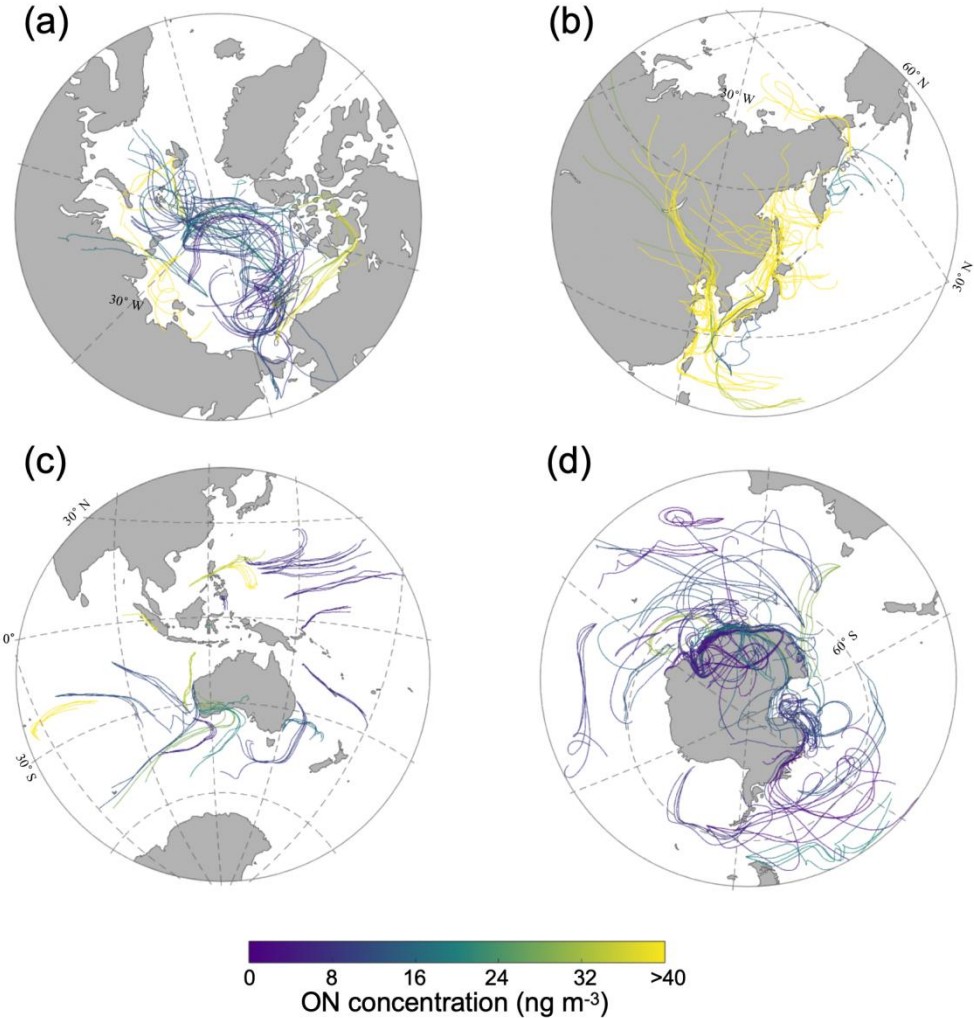

Figure 3. 5-day air-mass backward trajectories with ON concentrations and ON/TN ratios
along the Chinese Arctic/Antarctic expedition voyage over the Arctic Ocean (a), the coastal
East Asia (b), the Southeast Asia-Australia Tropical Ocean (c), and the Southern Ocean (d).
The SATO region exhibits intermediate level of ON concentrations (mean = 23.4
± 18.0 ng m$^{-3}$), lower than those influenced by anthropogenic activities in CEA but
higher than in polar regions. In this region, ON shows a significant positive
correlation with nssCa$^{2+}$ (Fig. 4; r = 0.76, $p < 0.01$), suggesting that terrestrial mineral
inputs (e.g., dust) influence ON levels, rather than purely marine sources. In addition,

backward trajectory analysis showed that samples affected by continental air masses have significantly higher ON concentrations than those exposed solely to marine air (Fig. 3c), suggesting the influences of continental sources. However, ON does not exhibit significant correlations with $nssK^+$ or with EC ($p > 0.05$), indicating that combustion emissions may not be the primary drivers. These findings suggest that the variability of ON in the SATO region results from a mixture of marine-terrestrial interactions, primarily modulated by episodic terrestrial mineral influence rather than continuous marine emissions. Notably, this region displays an elevated ON/TN ratio (Fig. 2), primarily due to its very low IN levels—approximately 85% lower than in the CEA region—which amplifies the relative contribution of ON within TN.

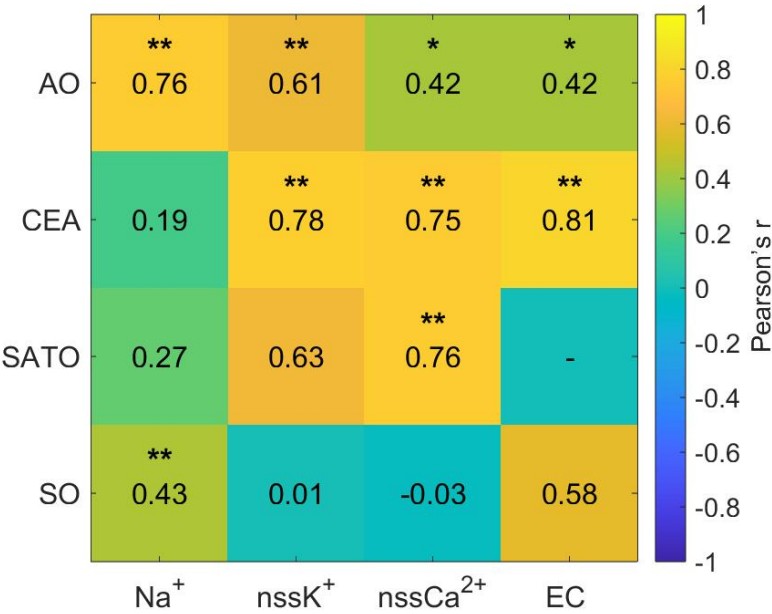

Figure 4. Spatial variations in the correlation coefficient between ON and chemical species across four regions (with "**" indicating $p < 0.01$ and "*" indicating $p < 0.05$).

In the AO region, ON concentrations were slightly lower than in the SATO region and significantly lower than in the CEA region. In this area, ON exhibited a significant positive correlation with $Na^+$, which suggest the sea salts inputs (Fig. 4; r = 0.43, $p < 0.01$), and also showed significant correlations with $nssK^+$ (r = 0.61, $p < 0.01$). Its correlations with $nssCa^{2+}$ (r = 0.42) and EC (r = 0.42) were weaker but still significant ($p < 0.05$). These patterns suggest that ON in the AO region may originate not only from primary sea-salt aerosols but may also be linked to biomass burning. However, the backward trajectory analysis shows no significant difference in ON concentrations between air masses influenced by continental sources and those

transported solely over the ocean (Fig. S1b; independent samples t-test, $p = 0.16$), likely suggesting the limited role of terrestrial inputs in this region. Unlike the SATO region, where ON showed no correlation with AEC (Fig. S2b; $p > 0.05$), ON concentrations in the AO exhibited a strong positive correlation with the AEC (Fig. S2a; $p < 0.01$), suggesting that marine biological activity is a key driver of ON variability in this region (Creamean et al., 2022). Collectively, these results demonstrate that the AO region is primarily governed by marine processes, with ON derived from both sea-spray organic enrichment and biogenic aerosol precursors, while terrestrial influences remain secondary (Nøjgaard et al., 2022).

In the SO region, ON concentrations were the lowest among all regions (mean = $12.0 \pm 7.1$ ng m$^{-3}$), yet the ON/TN ratio was relatively high ($27.8 \pm 11.0\%$). Back trajectory analysis indicates that air masses predominantly originated from the open ocean and Antarctic continent (Fig. 3d), with minimal anthropogenic influence. ON here exhibited a significant positive correlation with Na$^+$ (Fig. 4; $r = 0.43$, $p < 0.01$), but no significant relationships with nssK$^+$, nssCa$^{2+}$ or EC. While long-range transport events may deliver stable continental tracers like EC to this remote region, the lack of correlation between ON and these markers suggests that continental inputs are not the primary driver of ON variability. This pattern, combined with the association with Na$^+$ suggests that primary sea-salt emissions are an important pathway for ON in the SO atmosphere (Matsumoto et al., 2022), likely through the incorporation of marine-derived organic matter into sea-spray aerosols. Meanwhile, the absence of associations with terrestrial tracers further supports the notion that ON in this remote region is likely influenced more significantly by natural marine processes rather than continental or anthropogenic sources (Altieri et al., 2016).

**4.2 Role of sea-ice–associated biogenic processes in shaping Antarctic aerosol ON**

Sea-ice and open-ocean environments create distinct conditions for the production and emission of ON. While sea ice restricts direct air–sea exchange, it hosts specialized microbial communities and accumulates organic matter within brine channels. During melt and ice-edge retreat, this organic material is released into waters characterized by high primary productivity (Arrigo et al., 2008). This biological intensification enriches the surface microlayer and supplies precursors for aerosolization via sea spray and secondary formation (Dall'Osto et al., 2017; DeMott et al., 2016; Galgani et al., 2016; Wilson et al., 2015).

Along the Antarctic coast, we classified samples into two groups based on air-mass histories: open ocean (OO), influenced almost exclusively by open-ocean trajectories, and sea ice (SI), with air masses residing over sea ice for extended periods. SI samples exhibited significantly higher ON concentrations and ON/TN ratios than OO samples ($p < 0.001$; Mann–Whitney U test; Fig. 5a, b). Multiple lines of evidence point to sea-ice–associated biological processes as the driver of these enhancements: (1) Strong positive correlations of ON with sea-ice concentration (SIC; $r = 0.86$, $p < 0.01$) and with air-mass exposure to chlorophyll-a (AEC; $r = 0.91$, $p < 0.01$) in the SI group indicate that both ice cover and associated biological activity elevate ON (Fig. 5c, d); (2) PSCF analysis identifies high-probability source regions (PSCF > 0.8) over sea ice and its marginal zone for SI samples (Fig. S3c), consistent with an ice-edge origin; and (3) In contrast, ON shows no significant correlation with $Na^+$ ($r = -0.22$, $p > 0.05$) or with IN ($p > 0.05$) for SI samples (Fig. S4a,b), suggesting that primary sea-salt emissions and purely abiotic inorganic pathways are not the dominant contributors.

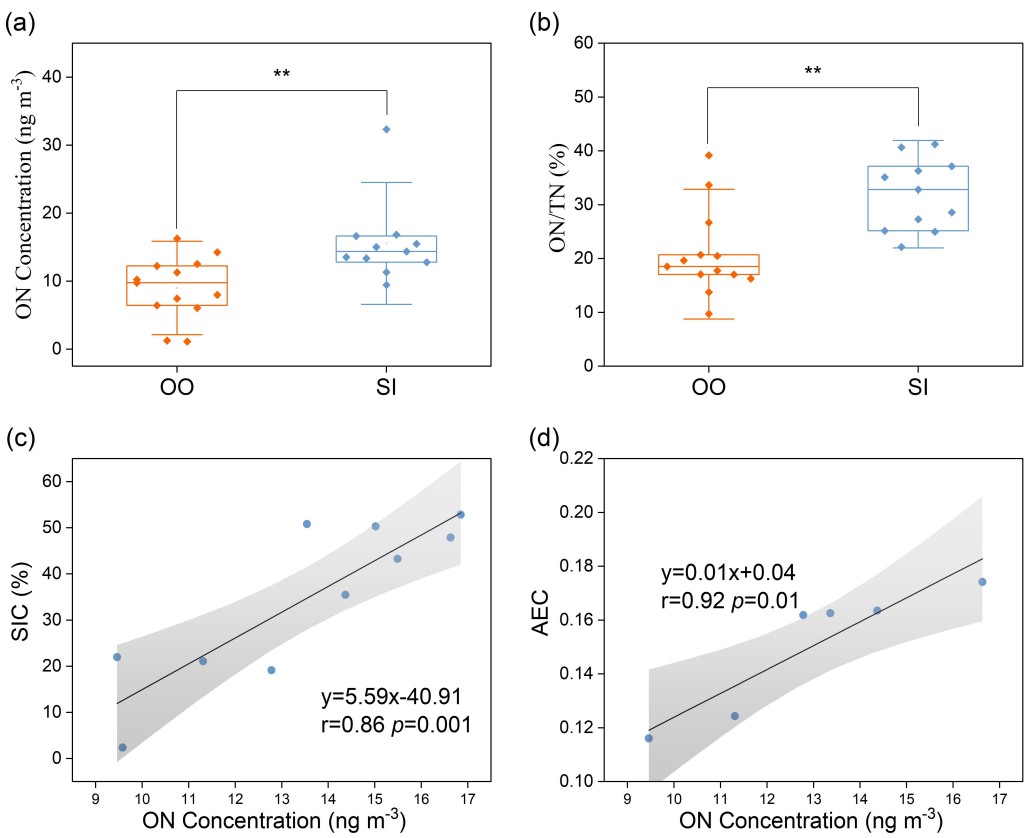

Figure 5. Comparison of measured ON concentrations (a) and ON/TN ratio (b) between SI and OO aerosol samples ("**" indicating $p < 0.01$). And correlations between SIC (c), AEC

(d) and ON concentration in SI aerosol samples. The sample sizes are n = 10 for panel (c) and
n = 6 for panel (d). These reduced sample sizes are due to unavailable satellite SIC/Chl-a data
along the trajectories, and in this study SIC or Chl-a is used only when $\geq$ 75% of the
trajectory points have valid satellite values (see Sections 2.6 – 2.7).
These observations support a mechanistic pathway whereby organic matter
released from sympagic (ice-associated) communities during melt enriches the surface
microlayer and is transferred to the atmosphere via sea spray as ON-rich particles
(DeMott et al., 2016; Wilson et al., 2015). Concurrently, a portion of this organic
nitrogen is rapidly microbially degraded to volatile alkylamines (e.g., methylamines)
(Taubert et al., 2017), which then form aminium salts through acid–base reactions
with marine emissions-derived acids (e.g., $H_2SO_4$, MSA), contributing to both ON
and IN in SI conditions (Brean et al., 2021; Dawson et al., 2012; Fitzsimons et al.,
2023). This process results in the formation of both organic (amine salts, contributing
to ON) and inorganic nitrogen aerosol species ($NH_4^+$ and $NO_3^-$), which explains their
elevated levels in the SI group samples (Fig. S5). The elevated ON/TN ratios in SI
samples (31.0%) relative to OO samples (20.8%) further indicate a greater fractional
contribution of ON under sea-ice influence (Fig. 5b), consistent with reported releases
of organic species from the sympagic ecosystem during melt (Jang et al., 2023;
Mirrielees et al., 2024; Yan et al., 2020).
For OO samples, PSCF hotspots (PSCF > 0.8) shift toward the offshore Southern
Ocean (Fig. S3d), in line with trajectories dominated by open-ocean air masses. The
positive association between ON and oceanic residence time (r = 0.66, p < 0.01; Fig.
S6) suggests that, as sea-ice influence diminishes, ON variability becomes
increasingly governed by open-ocean biological processes and long-range marine
aerosol transport.
Overall, these results establish the ice-edge/sympagic environment as an
important regulator of Antarctic aerosol ON. Sea-ice dynamics modulate both the
magnitude (higher ON and ON/TN) and sources (biogenic enrichment and
amine-driven secondary formation) of ON, underscoring the need to represent
sea-ice–associated processes in polar atmospheric chemistry and climate models.
**5. Conclusions and Implications**
Taking advantage of a new analytical tool for ON and aerosol samples collected from

three Antarctic and Arctic expeditions from 2019 to 2024, we quantified aerosol ON and IN in 92 TSP samples spanning 160° of latitude in the MABL. This dataset provides the first direct, subtraction-free ON measurements along a global-scale marine transect, capturing both water-soluble and water-insoluble fractions.

We observed a pronounced hemispheric and latitudinal gradient in ON, with substantially higher concentrations in the Northern Hemisphere ($83.3 \pm 141.4$ ng m$^{-3}$) than in the Southern Hemisphere ($15.4 \pm 12.4$ ng m$^{-3}$). Regionally, Coastal East Asia exhibited the highest ON ($164.6 \pm 179.1$ ng m$^{-3}$) but a low ON/TN ratio (21.1%), consistent with strong terrestrial and anthropogenic influences that elevate IN. The Southeast Asia–Australia Tropical Ocean showed intermediate ON and a relatively high ON/TN ratio due to low IN. The Arctic Ocean had lower ON but the highest ON/TN ratio (38.6%), indicating prominent marine biogenic contributions. The Southern Ocean showed the lowest ON ($12.0 \pm 7.0$ ng m$^{-3}$) yet a relatively high ON/TN ratio (27.8%), also suggestive of oceanic sources. Interannual variability across the three Antarctic campaigns was minor.

Multiple lines of evidence, including correlations with tracers, back-trajectory analysis, and PSCF, indicate that ON in CEA is dominated by continental inputs from combustion and dust, whereas ON in AO and SO is primarily controlled by marine processes. Along the Antarctic coast, air masses influenced by sea ice exhibited significantly higher ON and ON/TN than those influenced by the open ocean, with strong positive relationships to sea-ice concentration and air-mass exposure to chlorophyll-a. These patterns point to sympagic and ice-edge biogenic activity—through organic enrichment of sea spray and amine-driven secondary formation—as key regulators of ON near Antarctica.

Comparison with prior WSON-only datasets suggests that earlier studies likely underestimated total ON—by approximately 40% in the Southern Ocean—due to omission of WION. Accounting for both soluble and insoluble phases is therefore essential for constraining nitrogen deposition to the oceans and for representing ON's roles in atmospheric processes. Specifically, given that WION may significantly influence cloud condensation nuclei activity and cloud droplet formation, overlooking this fraction could lead to substantial uncertainties in assessing the radiative forcing and climate effects of marine aerosols.

These findings fill a critical observational gap, establish robust hemispheric and regional patterns of marine aerosol ON, and provide essential constraints for

atmospheric chemistry and climate models. Future efforts should explicitly represent ON sources, including sea-ice–associated biogenic processes and amine chemistry, and expand year-round, size-resolved, and composition-resolved measurements paired with isotopic and molecular tracers to refine source apportionment and evaluate model parameterizations across regions and seasons.

**Data availability.**

The data on organic nitrogen concentrations in aerosol are available at National Tibetan Plateau/Third Pole Environment Data Center, https://cstr.cn/18406.11.Atmos.tpdc.303043. DOI: https://doi.org/10.11888/Atmos.tpdc.303043 (Sun, 2025) [Dataset].

**Author contribution.**

Ningning Sun: Data curation, Writing-original draft. Yu Xu: Methodology. Bo Zhang and Ye Hu: Visualization, Software. Zhe Li: Methodology. Yilan Li: Carried out data analysis. Zhenlou Chen: Review. Jian Zhen Yu: Supervision, Writing – review & editing. Guitao Shi: Supervision, Writing – review & editing.

**Competing interests.**

The authors declare that they have no conflict of interest.

**Financial support**

This work was supported by the National Science Foundation of China (Grant Nos. 42276243 and 41922046), the Fundamental Research Funds for the Central Universities, the Hong Kong Research Grants Council (1621322, 16304924 and CRS_HKUST605/24)

**Acknowledgments**

The authors are grateful to CHINARE members for their support and assistance in sampling. The authors acknowledge the availability of the Hybrid Single-Particle Lagrangian Integrated Trajectory (HYSPLIT) model (available at https://www.arl.noaa.gov/hysplit/)

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

Index. (G02135, Version 3). [Data Set]. Boulder, Colorado USA. National Snow
and Ice Data Center. https://doi.org/10.7265/N5K072F8, 2017.
Fisher, J. A., Jacob, D. J., Travis, K. R., Kim, P. S., Marais, E. A., Chan Miller, C., Yu,
K., Zhu, L., Yantosca, R. M., Sulprizio, M. P., Mao, J., Wennberg, P. O., Crounse,
J. D., Teng, A. P., Nguyen, T. B., St. Clair, J. M., Cohen, R. C., Romer, P., Nault,
B. A., Wooldridge, P. J., Jimenez, J. L., Campuzano-Jost, P., Day, D. A., Hu, W.,
Shepson, P. B., Xiong, F., Blake, D. R., Goldstein, A. H., Misztal, P. K., Hanisco,
T. F., Wolfe, G. M., Ryerson, T. B., Wisthaler, A., and Mikoviny, T.: Organic
nitrate chemistry and its implications for nitrogen budgets in an isoprene- and
monoterpene-rich atmosphere: constraints from aircraft (SEAC[4] RS) and

ground-based (SOAS) observations in the Southeast US, Atmos. Chem. Phys., 16, 5969–5991, https://doi.org/10.5194/acp-16-5969-2016, 2016.

Fitzsimons, M. F., Tilley, M., and Cree, C. H. L.: The determination of volatile amines in aquatic marine systems: A review, Anal. Chim. Acta, 1241, 340707, https://doi.org/10.1016/j.aca.2022.340707, 2023.

Galgani, L., Piontek, J., and Engel, A.: Biopolymers form a gelatinous microlayer at the air-sea interface when Arctic sea ice melts, Sci. Rep., 6, 29465, https://doi.org/10.1038/srep29465, 2016.

Jang, J., Park, J., Park, J., Yoon, Y. J., Dall'Osto, M., Park, K.-T., Jang, E., Lee, J. Y., Cho, K. H., and Lee, B. Y.: Ocean-atmosphere interactions: Different organic components across Pacific and Southern Oceans, Sci. Total Environ., 878, 162969, https://doi.org/10.1016/j.scitotenv.2023.162969, 2023.

Jickells, T., Baker, A. R., Cape, J. N., Cornell, S. E., and Nemitz, E.: The cycling of organic nitrogen through the atmosphere, Phil. Trans. R. Soc. B, 368, 20130115, https://doi.org/10.1098/rstb.2013.0115, 2013.

Jung, J., Han, B., Rodriguez, B., Miyazaki, Y., Chung, H. Y., Kim, K., Choi, J.-O., Park, K., Kim, I.-N., Kim, S., Yang, E. J., and Kang, S.-H.: Atmospheric Dry Deposition of Water-Soluble Nitrogen to the Subarctic Western North Pacific Ocean during Summer, Atmosphere, 10, 351, https://doi.org/10.3390/atmos10070351, 2019.

Keene, W. C., Pszenny, A. A. P., Galloway, J. N., and Hawley, M. E.: Sea-salt corrections and interpretation of constituent ratios in marine precipitation, J. Geophys. Res., 91, 6647–6658, https://doi.org/10.1029/JD091iD06p06647, 1986.

Lesworth, T., Baker, A. R., and Jickells, T.: Aerosol organic nitrogen over the remote Atlantic Ocean, Atmos. Environ., 44, 1887–1893, https://doi.org/10.1016/j.atmosenv.2010.02.021, 2010.

Li, Y., Fu, T.-M., Yu, J. Z., Yu, X., Chen, Q., Miao, R., Zhou, Y., Zhang, A., Ye, J., Yang, X., Tao, S., Liu, H., and Yao, W.: Dissecting the contributions of organic nitrogen aerosols to global atmospheric nitrogen deposition and implications for ecosystems, Natl. Sci. Rev., 10, nwad244, https://doi.org/10.1093/nsr/nwad244, 2023.

Li, Y., Fu, T.-M., Yu, J. Z., Zhang, A., Yu, X., Ye, J., Zhu, L., Shen, H., Wang, C., Yang, X., Tao, S., Chen, Q., Li, Y., Li, L., Che, H., and Heald, C. L.: Nitrogen dominates global atmospheric organic aerosol absorption, Science, 387, 989–995,

https://doi.org/10.1126/science.adr4473, 2025.

Luo, L., Kao, S.-J., Bao, H., Xiao, H., Xiao, H., Yao, X., Gao, H., Li, J., and Lu, Y.: Sources of reactive nitrogen in marine aerosol over the Northwest Pacific Ocean in spring, Atmos. Chem. Phys., 18, 6207–6222, https://doi.org/10.5194/acp-18-6207-2018, 2018.

Mace, K. A., Duce, R. A., and Tindale, N. W.: Organic nitrogen in rain and aerosol at Cape Grim, Tasmania, Australia, J. Geophys. Res., 108, 2002JD003051, https://doi.org/10.1029/2002JD003051, 2003.

Matsumoto, K., Yamamoto, Y., Nishizawa, K., Kaneyasu, N., Irino, T., and Yoshikawa-Inoue, H.: Origin of the water-soluble organic nitrogen in the maritime aerosol, Atmos. Environ., 167, 97–103, https://doi.org/10.1016/j.atmosenv.2017.07.050, 2017.

Matsumoto, K., Kobayashi, H., Hara, K., Ishino, S., and Hayashi, M.: Water-soluble organic nitrogen in fine aerosols over the Southern Ocean, Atmos. Environ., 287, 119287, https://doi.org/10.1016/j.atmosenv.2022.119287, 2022.

Mirrielees, J. A., Kirpes, R. M., Costa, E. J., Porter, G. C. E., Murray, B. J., Lata, N. N., Boschi, V., China, S., Grannas, A. M., Ault, A. P., Matrai, P. A., and Pratt, K. A.: Marine aerosol generation experiments in the High Arctic during summertime, Elem Sci Anth, 12, 00134, https://doi.org/10.1525/elementa.2023.00134, 2024.

Miyazaki, Y., Kawamura, K., Jung, J., Furutani, H., and Uematsu, M.: Latitudinal distributions of organic nitrogen and organic carbon in marine aerosols over the western North Pacific, Atmos. Chem. Phys., 11, 3037–3049, https://doi.org/10.5194/acp-11-3037-2011, 2011a.

Miyazaki, Y., Kawamura, K., Jung, J., Furutani, H., and Uematsu, M.: Latitudinal distributions of organic nitrogen and organic carbon in marine aerosols over the western North Pacific, Atmos. Chem. Phys., 11, 3037–3049, https://doi.org/10.5194/acp-11-3037-2011, 2011b.

Ng, N. L., Brown, S. S., Archibald, A. T., Atlas, E., Cohen, R. C., Crowley, J. N., Day, D. A., Donahue, N. M., Fry, J. L., Fuchs, H., Griffin, R. J., Guzman, M. I., Herrmann, H., Hodzic, A., Iinuma, Y., Jimenez, J. L., Kiendler-Scharr, A., Lee, B. H., Luecken, D. J., Mao, J., McLaren, R., Mutzel, A., Osthoff, H. D., Ouyang, B., Picquet-Varrault, B., Platt, U., Pye, H. O. T., Rudich, Y., Schwantes, R. H., Shiraiwa, M., Stutz, J., Thornton, J. A., Tilgner, A., Williams, B. J., and Zaveri,

R. A.: Nitrate radicals and biogenic volatile organic compounds: oxidation, mechanisms, and organic aerosol, Atmos. Chem. Phys., 17, 2103–2162, https://doi.org/10.5194/acp-17-2103-2017, 2017.

Nøjgaard, J. K., Peker, L., Pernov, J. B., Johnson, M. S., Bossi, R., Massling, A., Lange, R., Nielsen, I. E., Prevot, A. S. H., Eriksson, A. C., Canonaco, F., and Skov, H.: A local marine source of atmospheric particles in the High Arctic, Atmos. Environ., 285, 119241, https://doi.org/10.1016/j.atmosenv.2022.119241, 2022.

Park, K., Lee, K., Kim, T., Yoon, Y. J., Jang, E., Jang, S., Lee, B., and Hermansen, O.: Atmospheric DMS in the Arctic Ocean and Its Relation to Phytoplankton Biomass, Global Biogeochemical Cy., 32, 351–359, https://doi.org/10.1002/2017GB005805, 2018.

Pavuluri, C. M., Kawamura, K., and Fu, P. Q.: Atmospheric chemistry of nitrogenous aerosols in northeastern Asia: biological sources and secondary formation, Atmos. Chem. Phys., 15, 9883–9896, https://doi.org/10.5194/acp-15-9883-2015, 2015.

Quinby-Hunt, M. S. and Turehian, K. K.: Distribution of elements in sea water, EoS Trans. Am. Geophys. Union, 64, 130–130, https://doi.org/10.1029/EO064i014p00130, 1983.

Shi, G., Ma, H., Zhu, Z., Hu, Z., Chen, Z., Jiang, S., An, C., Yu, J., Ma, T., Li, Y., Sun, B., and Hastings, M. G.: Using stable isotopes to distinguish atmospheric nitrate production and its contribution to the surface ocean across hemispheres, Earth Planet. Sci. Lett., 564, 116914, https://doi.org/10.1016/j.epsl.2021.116914, 2021.

Shubhankar, B. and Ambade, B.: Chemical characterization of carbonaceous carbon from industrial and semi urban site of eastern India, SpringerPlus, 5, 837, https://doi.org/10.1186/s40064-016-2506-9, 2016.

Siegel, D. A., Behrenfeld, M. J., Maritorena, S., McClain, C. R., Antoine, D., Bailey, S. W., Bontempi, P. S., Boss, E. S., Dierssen, H. M., Doney, S. C., Eplee, R. E., Evans, R. H., Feldman, G. C., Fields, E., Franz, B. A., Kuring, N. A., Mengelt, C., Nelson, N. B., Patt, F. S., Robinson, W. D., Sarmiento, J. L., Swan, C. M., Werdell, P. J., Westberry, T. K., Wilding, J. G., and Yoder, J. A.: Regional to global assessments of phytoplankton dynamics from the SeaWiFS mission, Remote Sens. Environ., 135, 77–91, https://doi.org/10.1016/j.rse.2013.03.025,

800     2013.

Song, J., Zhao, Y., Zhang, Y., Fu, P., Zheng, L., Yuan, Q., Wang, S., Huang, X., Xu,
W., Cao, Z., Gromov, S., and Lai, S.: Influence of biomass burning on
atmospheric aerosols over the western South China Sea: Insights from ions,
carbonaceous fractions and stable carbon isotope ratios, Environ. Pollut., 242,
1800–1809, https://doi.org/10.1016/j.envpol.2018.07.088, 2018.
Stein, A. F., Draxler, R. R., Rolph, G. D., Stunder, B. J. B., Cohen, M. D., and Ngan,
F.: NOAA's HYSPLIT Atmospheric Transport and Dispersion Modeling System,
Bull. Am. Meteorol. Soc., 96, 2059–2077,
https://doi.org/10.1175/BAMS-D-14-00110.1, 2015.
Sun, N.: Spatial distribution of organic nitrogen in the global marine boundary layer
from Arctic to Antarctic, National Tibetan Plateau / Third Pole Environment
Data Center), https://doi.org/10.11888/Atmos.tpdc.303043, 2025.
Taubert, M., Grob, C., Howat, A. M., Burns, O. J., Pratscher, J., Jehmlich, N., Von
Bergen, M., Richnow, H. H., Chen, Y., and Murrell, J. C.: Methylamine as a
nitrogen source for microorganisms from a coastal marine environment, Environ.
Microbiol., 19, 2246–2257, https://doi.org/10.1111/1462-2920.13709, 2017.
Tian, M., Li, H., Wang, G., Fu, M., Qin, X., Lu, D., Liu, C., Zhu, Y., Luo, X., Deng,
C., Abdullaev, S. F., and Huang, K.: Seasonal source identification and
formation processes of marine particulate water soluble organic nitrogen over an
offshore island in the East China Sea, Sci. Total Environ., 863, 160895,
https://doi.org/10.1016/j.scitotenv.2022.160895, 2023.
Violaki, K., Sciare, J., Williams, J., Baker, A. R., Martino, M., and Mihalopoulos, N.:
Atmospheric water-soluble organic nitrogen (WSON) over marine environments:
a global perspective, Biogeosciences, 12, 3131–3140,
https://doi.org/10.5194/bg-12-3131-2015, 2015a.
Violaki, K., Sciare, J., Williams, J., Baker, A. R., Martino, M., and Mihalopoulos, N.:
Atmospheric water-soluble organic nitrogen (WSON) over marine environments:
a global perspective, Biogeosciences, 12, 3131–3140,
https://doi.org/10.5194/bg-12-3131-2015, 2015b.
Wilson, T. W., Ladino, L. A., Alpert, P. A., Breckels, M. N., Brooks, I. M., Browse, J.,
Burrows, S. M., Carslaw, K. S., Huffman, J. A., Judd, C., Kilthau, W. P., Mason,
R. H., McFiggans, G., Miller, L. A., Nájera, J. J., Polishchuk, E., Rae, S.,
Schiller, C. L., Si, M., Temprado, J. V., Whale, T. F., Wong, J. P. S., Wurl, O.,
Yakobi-Hancock, J. D., Abbatt, J. P. D., Aller, J. Y., Bertram, A. K., Knopf, D.
A., and Murray, B. J.: A marine biogenic source of atmospheric ice-nucleating
particles, Nature, 525, 234–238, https://doi.org/10.1038/nature14986, 2015.
Wu, C. and Yu, J. Z.: Determination of primary combustion source organic
carbon-to-elemental carbon (OC / EC) ratio using ambient OC and EC
measurements: secondary OC-EC correlation minimization method, Atmos.
Chem. Phys., 16, 5453–5465, https://doi.org/10.5194/acp-16-5453-2016, 2016.
Wu, G., Hu, Y., Gong, C., Wang, D., Zhang, F., Herath, I. K., Chen, Z., and Shi, G.:
Spatial distribution, sources, and direct radiative effect of carbonaceous aerosol
along a transect from the Arctic Ocean to Antarctica, Sci. Total Environ., 916,
170136, https://doi.org/10.1016/j.scitotenv.2024.170136, 2024.
Xiao, H.-W., Xiao, H.-Y., Luo, L., Shen, C.-Y., Long, A.-M., Chen, L., Long, Z.-H.,
and Li, D.-N.: Atmospheric aerosol compositions over the South China Sea:
temporal variability and source apportionment, Atmos. Chem. Phys., 17,
3199–3214, https://doi.org/10.5194/acp-17-3199-2017, 2017.
Yan, J., Jung, J., Lin, Q., Zhang, M., Xu, S., and Zhao, S.: Effect of sea ice retreat on
marine aerosol emissions in the Southern Ocean, Antarctica, Sci. Total Environ.,
745, 140773, https://doi.org/10.1016/j.scitotenv.2020.140773, 2020.
Yan, S., Xu, G., Zhang, H., Wang, J., Xu, F., Gao, X., Zhang, J., Wu, J., and Yang, G.:
Factors Controlling DMS Emission and Atmospheric Sulfate Aerosols in the
Western Pacific Continental Sea, J. Geophys. Res., 129, e2024JC020886,
https://doi.org/10.1029/2024JC020886, 2024.
Yu, X., Li, Q., Ge, Y., Li, Y., Liao, K., Huang, X. H., Li, J., and Yu, J. Z.:
Simultaneous Determination of Aerosol Inorganic and Organic Nitrogen by
Thermal Evolution and Chemiluminescence Detection, Environ. Sci. Technol.,
55, 11579–11589, https://doi.org/10.1021/acs.est.1c04876, 2021.
Zamora, L. M., Prospero, J. M., and Hansell, D. A.: Organic nitrogen in aerosols and
precipitation at Barbados and Miami: Implications regarding sources, transport
and deposition to the western subtropical North Atlantic, J. Geophys. Res., 116,
D20309, https://doi.org/10.1029/2011JD015660, 2011.
Zhou, S., Chen, Y., Paytan, A., Li, H., Wang, F., Zhu, Y., Yang, T., Zhang, Y., and
Zhang, R.: Non-Marine Sources Contribute to Aerosol Methanesulfonate Over
Coastal Seas, JGR Atmospheres, 126, e2021JD034960,
https://doi.org/10.1029/2021JD034960, 2021.
Zhou, S., Chen, Y., Wang, F., Bao, Y., Ding, X., and Xu, Z.: Assessing the Intensity

869       of Marine Biogenic Influence on the Lower Atmosphere: An Insight into the

870       Distribution of Marine Biogenic Aerosols over the Eastern China Seas, Environ.

871       Sci. Technol., 57, 12741–12751, https://doi.org/10.1021/acs.est.3c04382, 2023.