# Peer review of "Ningning Sun1,2, Xu Yu2,3, Jian Zhen Yu2,4\*, Bo Zhang1, Yilan Li1,5, Ye Hu1, Zhe Li1,"

_EGUsphere, 2025_

## Author Comment (AC1)

Referee: 1

We are grateful to reviewer#1 for the detailed comments and constructive suggestions, which have greatly improved the manuscript. Below we provide a point-by-point response. Referee comments are in black, and our responses are in blue.

This manuscript presents the first direct measurements of aerosol organic nitrogen (ON) along a global marine atmospheric boundary layer transect, revealing clear latitudinal and regional patterns. The study is based on substantial and carefully conducted field campaigns and addresses an important knowledge gap in marine atmospheric nitrogen cycling. The results are potentially impactful for understanding ON sources and for improving atmospheric and climate model representations. I therefore recommend publication of this manuscript after the issues and questions raised below are adequately addressed.

**Response**: We appreciate the Reviewer's positive comments on our manuscript. Below, we give a point-by-point response to your comments and suggestions.

Special comments:

(1) Line 33. Please check the unit "ng mm$^{-3}$" is right?

**Response**: Thank you for pointing this out. It has now been corrected to "ng m$^{-3}$" in the abstract.

(2) Line 35. The variance of ON/TN ratio should also be mentioned

**Response**: We agree. In the revised manuscript, we have added the statistical variability of the ON/TN ratio to better characterize its spatial variability along the transect. This has been added to both the Abstract and the Results sections.

We revised the main text as follows:

Abstract: "*A significant latitudinal gradient was observed, with significantly higher ON concentrations in the Northern Hemisphere (83.3 ± 141.4 ng m$^{-3}$) than in the Southern Hemisphere (15.4 ± 12.4 ng m$^{-3}$). Regionally, coastal East Asia recorded the highest ON levels (164.6 ± 179.1 ng m$^{-3}$) but a lower ON/TN ratio (21.1 ± 7.9%), indicating strong terrestrial and anthropogenic influence. In contrast, the Arctic Ocean had lower ON concentrations (19.1 ± 19.0 ng m$^{-3}$) but the highest ON/TN ratio (38.6 ± 12.4%), suggesting dominant marine biogenic sources. The Southern Ocean showed the lowest ON concentration (12.0 ± 7.1 ng m$^{-3}$) yet a relatively high ON/TN ratio (27.8 ± 11.0%), also pointing to oceanic origins.*"

Results: "*The CEA region exhibited the highest ON concentrations (mean = 164.6 ng m$^{-3}$) but the lowest ON/TN ratio (mean = 21.1 ± 7.9%). In contrast, the SO region showed the lowest ON concentrations (mean = 12.0 ng m$^{-3}$; range: 1.78–32.3 ng m$^{-3}$)*

*and higher ON/TN ratios (mean = 27.8 ± 11.0%). Notably, the AO region displayed the highest ON/TN ratios (mean = 38.6 ± 12.4%) despite relatively low ON concentrations (mean = 19.1 ng m⁻³; range: 5.2–32.2 ng m⁻³). The ON/TN ratio in SATO region (26.8 ± 10.0%) is similar to that of SO but with a lower ON concentration (mean = 23.4 ng m⁻³; range: 5.7–70.1 ng m⁻³), which is much lower than the CEA region, but higher than the high latitude two pole regions.*"

(3) Line 56-102. The narrative flow of the "Introduction" could be improved. For example, the first paragraph highlights the limitations of traditional ON analytical approaches but does not immediately introduce your solutions; and the discussion of ON sources in the second and fourth paragraphs is interrupted by the third paragraph on the geochemical significance of ON, which makes the overall structure somewhat difficult to follow.

**Response**: We thank the reviewer for this constructive suggestion. To improve the narrative flow, we restructured the Introduction. Specifically, we moved the key solution statement (simultaneous IN/ON measurement capturing both WSON and WION without subtraction) to the end of the first paragraph to immediately address the limitation raised. We also grouped all source-related discussion consecutively and moved the paragraph on the geochemical/climate significance after the source discussion so that the overall structure is easier to follow.

The revised Introduction is as follows:

[revised manuscript text omitted]

(4) Line 169-184. Given the very low ON concentrations in marine aerosol samples, it would be helpful for the authors to clarify the detection limit of the new method and whether it offers a clear advantage over traditional IC-based approaches? Additionally,

more information on the separation and discrimination between IN and ON would strengthen confidence in the measurements.

**Response**: The ON measurements in this study follow the thermal evolution–chemiluminescence method developed by Yu et al. (2021), which reports a method detection limit of 96 ng N (absolute mass on the analyzed filter aliquot). In practice, low ambient concentrations can be reliably measured by increasing the analyzed filter area. In this study, we typically used 4–6 cm$^2$ of filter per sample for analysis, which proportionally lowers the effective concentration detection limit and ensures reliable ON quantification.

Compared with traditional IC-based approaches, this method offers a clear advantage by enabling the simultaneous determination of IN and ON on the same filter aliquot, thereby avoiding the difference method (ON = TN − IN) and the associated large uncertainty propagation when TN and IN are similar in magnitude. IN/ON discrimination is based on their distinct thermal evolution behavior and the joint interpretation of C and N thermograms: ON is identified by co-evolving C and N signals across temperature steps, whereas IN is characterized by N-only evolution without a corresponding C signal. Overlapping features are further separated using multivariate curve resolution (MCR), as validated using laboratory standards, synthetic mixtures, and ambient aerosol samples in Yu et al. (2021).

We have added the method detection limit description, the effective detection limit used in this study, and a clearer explaination of the IN/ON separation and discrimination (C–N thermograms combined with MCR/PMF) to the Methods section, as follows:

"*Aerosol ON and IN were simultaneously measured using the recently developed Aerosol Nitrogen Analyzer system, which enables sensitive quantification directly from filter samples without pretreatment. Detailed descriptions of the method are provided in Yu et al (2021). Briefly, the method detection limit is 96 ng N. Because the detection limit scales inversely with the analyzed filter area, it can be readily lowered by analyzing a larger aliquot. In this study, 4–6 cm$^2$ of filter material was typically analyzed for each sample, yielding a proportionally lower effective detection limit and ensuring stable and reliable quantification for low-concentration marine aerosol samples. Compared with traditional IC-based approaches, this analyzer provides a clear advantage by determining IN and ON simultaneously on the same filter aliquot, thereby avoiding the subtraction-based "difference method" (ON = TN − IN) and the associated uncertainty propagation when TN and IN are similar in magnitude.*

*The analyzer integrates a thermal aerosol carbon analyzer and a chemiluminescence NOx analyzer. Aerosol samples collected on quartz fiber filters were thermally evolved under a programmed 6-step temperature protocol (150, 180, 300, 400, 500, and 800 °C) in a 1% O$_2$/99% He carrier gas. The evolved materials were catalytically oxidized to CO$_2$ and nitrogen oxides (NO$_y$), with the C signal monitored via methanator-FID detection and the N signal recorded through chemiluminescence*

*after converting NOy to NO. The C signal assists in differentiating IN and ON components, as ON aerosols produce both C and N signals while the IN fraction only yields an N signal. The programmed thermal evolution facilitates separation of aerosol IN and ON due to their distinct thermal characteristics. Specifically, IN and ON discrimination is achieved by jointly interpreting the C and N thermograms: ON is identified by co-evolving C and N signals across the temperature steps, whereas IN is characterized by N-only evolution without a corresponding C signal. The separation of overlapping thermal features is further resolved using multivariate curve resolution (MCR), which deconvolves the mixed thermograms into source-like components based on their distinct thermal evolution patterns. Quantification of IN and ON is achieved through multivariate curve resolution (MCR) data treatment of the C and N thermograms using USEPA PMF (version 5.0)."*

Reference:

Yu, X., Li, Q., Ge, Y., Li, Y., Liao, K., Huang, X. H., Li, J., and Yu, J. Z.: Simultaneous Determination of Aerosol Inorganic and Organic Nitrogen by Thermal Evolution and Chemiluminescence Detection, Environ. Sci. Technol., 55, 11579–11589, https://doi.org/10.1021/acs.est.1c04876, 2021.

(5) Line 191. I didn't get it. As the ship was continuously moving and each sample integrates air masses over approximately 2–4 degrees of latitude, it is not entirely clear how the backward air-mass trajectories were constructed for an individual sample. The authors should clarify how the temporal and spatial variability during sampling was accounted for in the trajectory analysis.

**Response**: We agree that, because the ship was continuously moving and each sample integrates air masses over ~2–4° of latitude, the trajectory analysis must account for spatiotemporal variability during the 48 h sampling period. We therefore implemented a two-step trajectory analysis strategy. First, for the entire dataset, a standard characterization was conducted using a representative midpoint approach (the average of the start and end coordinates) with trajectories generated at 6-hour intervals. This provides a consistent, large-scale overview of the air-mass history for all samples. However, we recognize that this simplified approach may not capture the fine-scale heterogeneity required for investigating sea-ice influences.

Accordingly, for the Southern Ocean and Antarctic marginal regions (Section 4.2), we conducted a high-resolution, moving-track strategy. For each 48 h sample, we divided the cruise track into 48 points (hourly positions) and calculated a 120 h (5-day) backward trajectory for each point. This high-resolution ensemble approach explicitly accounts for the ship's continuous movement and the spatial integration of the samples, providing a robust basis for calculating parameters such as sea-ice residence time and diagnostics. We have clarified this in the revised Methods text:

*"Air mass backward trajectories were simulated using the HY-SPLIT model with meteorological fields from the NOAA GDAS database to reveal the transport history of air masses arriving at the vessel (Stein et al., 2015). Given that the ship was continuously moving and each sample integrates air masses over approximately 2–4 degrees of latitude, we applied a nested strategy to account for spatiotemporal variability. For the initial characterization of the entire dataset, a representative sampling location was defined for each sample using the average latitude and longitude of its start and end positions, with backward trajectories simulated at 6-h intervals anchored to this midpoint to identify dominant air-mass categories (Fig. 3). Subsequently, to precisely investigate the influence of sea ice on ON in the Southern Ocean and Antarctic marginal regions (Section 4.2), a targeted high-resolution analysis was performed on this subset of samples. For each Antarctic sample, the actual cruise track was equally divided into 48 points corresponding to the hourly intervals of the 48-h sampling period, and a 120-h (5-day) backward trajectory was calculated for each of these 48 coordinates (Fig. S3a and b). "*

**Response**: We thank the reviewer for this valuable suggestion. Data normality was assessed using the Shapiro-Wilk test, and the results are provided in the Supporting Information (Table S1). Because several datasets deviated from normality; we used the non-parametric Mann-Whitney U tests. The statistical significance and overall conclusions are consistent with those obtained using t-tests, supporting the robustness of the results shown in Line 245 and Figure 5.

This point was clarified in the revised manuscript:

Section 3: "*Atmospheric ON concentrations exhibited significant hemispheric differences ($p < 0.001$; Mann–Whitney U test; Table S1), with values in the Northern Hemisphere (NH: $83.3 \pm 141.4$ ng m$^{-3}$, $N = 55$) being approximately five times higher than those in the Southern Hemisphere (SH: $15.4 \pm 12.4$ ng m$^{-3}$, $N = 37$).*"

Section 4.2: "*Along the Antarctic coast, we classified samples into two groups based on air-mass histories: open ocean (OO), influenced almost exclusively by open-ocean trajectories, and sea ice (SI), with air masses residing over sea ice for extended periods. SI samples exhibited significantly higher ON concentrations and ON/TN ratios than OO samples (; $p < 0.001$; Mann–Whitney U test; Fig. 5a, b).*"

*Table S1. Normality assessment and non-parametric statistical tests for comparisons*

| Comparison | Group A (n) | Group B (n) | Shapiro–Wilk p (A) | Shapiro–Wilk p (B) | Test applied | p-value |
|---|---|---|---|---|---|---|
| ON (NH vs SH) | 51 | 40 | $2.27 \times 10^{-11}$ | $1.02 \times 10^{-6}$ | Mann–Whitney U | $8.43 \times 10^{-4}$ |
| ON (Sea-ice vs non–sea-ice) | 13 | 11 | 0.585 | $7.58 \times 10^{-4}$ | Mann–Whitney U | $2.01 \times 10^{-3}$ |
| ON/TN (Sea-ice vs non–sea-ice) | 13 | 11 | 0.0107 | 0.69 | Mann–Whitney U | $9.01 \times 10^{-4}$ |

(8). Line 276. I am confused about how total nitrogen (TN) was determined using ion chromatography in previous studies (Table 1); could the authors clarify whether the reported WSON/TN ratios in the literature actually refer to WSON/WSTN instead?

**Response**: We appreciate this important comment. In the studies summarized in Table 1, "TN" was determined on aqueous extracts (e.g., via persulfate oxidation, UV oxidation, or TOC/TN analyzer methods), and therefore represents **water-soluble total nitrogen (WSTN)** rather than total aerosol nitrogen. Accordingly, the ON/TN ratios reported in the literature should be interpreted as WSON/WSTN rather than total aerosol nitrogen. This point has now been clarified in Table 1.

(9) Table1: I suggest to add your own dataset in Table1

**Response**: Thanks for this suggestion. We have now added our dataset to Table 1. This inclusion facilitates direct comparison between our global marine transect observations and previously reported studies.

The modified table is as follows:

Table 1. Measured concentrations of WSON from published reports in the marine atmospheric boundary layer.

| Regions | Period | Locations | WSON (ng m⁻³) | Ratio to WSTN(%) | Methods for IN | Methods for WSTN | Reference |
|---|---|---|---|---|---|---|---|
| Southern Atlantic | 2007.01-02 | 35° S–45° S | $119 \pm 163.8$ | - | IC | PO | (Violaki et al., 2015b) |
| the Northwest Pacific Ocean | 2014.2015 | 25–40° N, 125–150° E | 43.4–564.2 | 11–46 | IC | PO | (Luo et al., 2018) |
| the coast of China the East China seas | 2014.2015 | 30–40° N, 120–130° E | 96.6–7238 | 6–48 | IC | PO | (Luo et al., 2018) |
| Southern Ocean | 2000.11 | 40.41° S, 144.41° E | $50.4 \pm 79.8$ | 21 | IC | UV | (Mace et al., 2003) |
| Oahu, Hawaii | 1998.7.22-8.13 | 21.7° N, 157.8° W | 46.2 | 31 | IC | UV | (Cornell et al., 2001) |
| the southern margin of the East China Sea | 2005-2006 | 25.09° N, 121.46° E | $476 \pm 756$ | $24 \pm 16$ | IC | UV with PO | (Chen and Chen, 2010) |
| the western North Pacific | 2008.08.24-09.13 | 42.98° N,144.37° E –35.65° N, 139.77° E | $130 \pm 61$ (10–260) | $67 \pm 15$ | IC | TOC/TN analyzer | (Miyazaki et al., 2011b) |
| Huaniao Island | 2019 | 30.86° N, 122.67° E | 30–2810 | 0.13–77 | IC | TOC/TN analyzer | (Tian et al., 2023) |
| the northern tip of Japan | 2010-2012 | ~45.2° N, ~141.2° E | $77 \pm 57$ | $12.8 \pm 15.2$ | IC | TOC/TN analyzer | (Matsumoto et al., 2017) |
| the Southern Ocean | 2016-2020 | 38.8–69.0° S, 38.1–150.8° E | 4.7 | 20 | IC | TOC/TN analyzer | (Matsumoto et al., 2022) |
| the Subarctic Western North Pacific Ocean | 2016.7.21-8.22 | 30–65° N, 130–160° W | 1.62–205.8 | 9 | IC | TOC/TN analyzer | (Jung et al., 2019) |
| Bermuda | 2011 | 32.27° N, 64.87° W | $105 \pm 191.8$ | $50.4 \pm 18.9$ | nutrient analyzer | TN Analyzer | (Altieri et al., 2016) |
| the Arctic Ocean | 2021.07-09 2021.07-09, | north of ~60° N | 5.2–32.2 | $38.6 \pm 12.4$ | N Analyzer | N Analyzer | this study |
| the Coastal East Asia | 2019.10-11, 2023.10-2024.04 | 20–60° N | 18.1–555.6 | $21.1 \pm 7.9$ | N Analyzer | N Analyzer | this study |
| Athe Southeast Asia-Australia Tropical Ocean | 2021.11, 2023.11 | 20° N-40° S | 5.7–70.1 | $26.8 \pm 10.0$ | N Analyzer | N Analyzer | this study |
| the Southern Ocean | 2021.11-2022.03, 2019.11, 2023.11 | south of ~40° S | 1.8–32.3 | $27.8 \pm 11.0$ | N Analyzer | N Analyzer | this study |

*PO: the persulfate oxidation (PO) method

*UV: ultraviolet photo-oxidation

*TN analyzer: a total organic carbon (TOC) analyzer with a TN unit

*Nutrient analyzer: automated nutrient analyzer and standard colorimetric method

(10) Line 295 "4.1 Source identification of ON". As correlation alone does not allow one to distinguish between common emission sources and shared atmospheric transport or processing, I suggest that the authors either temper causal language (e.g., "dominated by", "primarily controlled by") or provide additional independent evidence to strengthen the source attribution.

**Response**: We agree that correlation alone cannot unambiguously separate common emission sources from shared transport and/or atmospheric processing. Our source interpretation is not based on correlations alone. We also consider (i) air-mass origin and residence-time diagnostics from backward trajectories (marine/continental/sea-ice/Antarctic influence), (ii) consistency with established source tracers (e.g., $Na^+$ for sea-salt/sea-spray influence, $nssCa^{2+}$ for crustal/dust inputs, $nssK^+$ for biomass burning, EC for combustion-related influence), (iii) contrasts between air-mass categories (e.g., continental-influenced vs. marine-only) supported by statistical tests, and (iv) marine biological proxies (AEC derived from Chl-a along trajectories). These complementary constraints help identify the most plausible dominant contributors to ON variability in each region.

Nevertheless, to avoid over-interpreting correlations, we revised Section 4.1 to temper causal wording (e.g., replacing "dominated by/primarily controlled by" with "suggests/consistent with/likely influenced by") and to explicitly reference the multiple lines of evidence used.

The revised text is provided below:

"*A significant correlation between ON and the anthropogenic tracer EC (r = 0.81, p < 0.01; Fig. 4) indicates that fossil fuel combustion and biomass burning are important ON sources .*"
"* Meanwhile, the absence of associations with terrestrial tracers further supports the notion that ON in this remote region is likely influenced more significantly by natural marine processes rather than continental or anthropogenic sources.*"
"*This pattern, combined with the association with Na$^+$ suggests that primary sea-salt emissions are an important pathway for ON in the SO atmosphere (Matsumoto et al., 2022), likely through the incorporation of marine-derived organic matter into sea-spray aerosols.*"

(11) Line 327-335. The author said that the correlation between ON and $nss\text{-}Ca^{2+}$ is good, but there is no correlation between ON and $nss\text{-}K^+$ or EC. Is this a contradiction? I suggest the authors further explain this inconsistency. From the perspective of air masses, the backward trajectories of SATO samples have few intersections with the continental region. Therefore, can $nss\text{-}Ca^{2+}$ be regarded as a reliable indicator of continental transport?

**Response**: This is a very insightful question. We agree that the different behavior of $nssCa^{2+}$ versus $nssK^+$ and EC in the SATO region requires clarification. For the SATO samples, the air-mass origins can be divided into two types: (1) trajectories that

traveled only over the ocean and (2) trajectories that passed over land (Fig. R1). Figure S1a shows a significant difference between these two groups. The ON concentrations are significantly higher in the land-influenced samples than in the ocean-only samples. Specifically, although some air masses passed over land (e.g., Southeast Asian islands and coastal Australia), these regions do not possess the same high-intensity emission profiles as the East Asian continent.

In this context, $nssCa^{2+}$ serves as a tracer of mineral/dust from the continents, which will affect remote marine air via long-range transport in the free troposphere followed by subsidence and mixing into the marine boundary layer. Different from $nssCa^{2+}$, $nssK^+$ and EC are more closely associated with biomass burning/combustion emissions. The lack of ON–$nssK^+$ and ON–EC correlations suggests that combustion-related influence is limited and not the dominant driver of ON variability in this subset, whereas the ON–$nssCa^{2+}$ relationship points to episodic mineral influence or other non-combustion terrestrial sources (e.g., soil organic matter associated with dust, terrestrial biogenic particles, or agricultural emissions). We have revised the text to clarify these points and to avoid treating $nssCa^{2+}$ as an exclusive proxy for continental transport, as follows:

"*The SATO region exhibits intermediate level of ON concentrations (mean = 23.4 ± 18.0 ng m$^{-3}$), lower than those influenced by anthropogenic activities in CEA but higher than in polar regions. In this region, ON shows a significant positive correlation with $nssCa^{2+}$ (Fig. 4; r = 0.76, p < 0.01), suggesting that terrestrial mineral inputs (e.g., dust) influence ON levels, rather than purely marine sources. In addition, backward trajectory analysis showed that samples affected by continental air masses have significantly higher ON concentrations than those exposed solely to marine air (Fig. 3c), suggesting the influences of continental sources. However, ON does not exhibit significant correlations with $nssK^+$ or with EC (p > 0.05), indicating that combustion emissions may not be the dominant driver in this region. These findings suggest that the variability of ON in the SATO region results from a mixture of marine-terrestrial interactions, primarily modulated by episodic terrestrial mineral influence rather than continuous marine emissions. Notably, this region displays an elevated ON/TN ratio (Fig. 2), primarily due to its very low IN levels—approximately 85% lower than in the CEA region—which amplifies the relative contribution of ON within TN.*"

[Figure]

**Figure R1.** Analysis of backward trajectory for the samples collected in the Southeast Asia-Australia Tropical Ocean region on November 10, 2021 (a), November 13, 2021 (b), November 20, 2021 (c), November 10, 2023 (d), November 12, 2023(e) and November 14, 2023 (f).

End of responses to the Referee.

---

## Author Comment (AC2)

Referee: 2

We thank Reviewer#2 for detailed comments and constructive suggestions, which have greatly improved the manuscript. Below, we provide a point-by-point response to the comments. Referee comments are in black, and the responses are in blue.

Sun et al. reported measurements of organic nitrogen (ON) across the global marine atmospheric boundary layer. They observed that ON/TN ratios in polar regions are significantly higher than in other regions, attributing this difference to marine biogenic emissions. Furthermore, in sea ice-covered areas of Antarctica, elevated ON concentrations were detected, with sea-ice-associated ecosystems proposed as the primary driver for enhanced ON production. Overall, this manuscript presents the first global-scale measurements of ON and provides measurement-based ON/TN ratios for the marine boundary layer, thereby offering a very valuable dataset and novel insights into atmospheric ON.

Response: We thank the reviewer for the positive assessment of our work. We have carefully addressed all comments and revised the manuscript accordingly to improve methodological clarity and interpretation.

Comment 1: One major concern is the potential lack of methodological details. Specifically, how did the authors calculate sea ice concentrations for individual samples? While remote sensing data are mentioned for illustrating spatiotemporal distributions of sea ice in the Southern Ocean, the types of remote sensing data employed are not specified, indicating a need for further clarification. Also see my comments below.

Response: Thank you for raising this important point. Sea ice concentrations (SIC) for individual samples were calculated using satellite-derived daily gridded passive microwave SIC products. For regional-scale visualization of sea-ice extent (SIE) and SIC variability, we used the Sea Ice Index (Version 3) distributed by NSIDC (Fetterer et al., 2017), derived from passive-microwave observations (SSM/I – SSMIS). For sample-level calculations, SIC values along trajectories were obtained from the AMSR2 daily gridded SIC product (University of Bremen, ASI algorithm; Version 5.4). For each trajectory endpoint, SIC was extracted by collocating the endpoint latitude/longitude and date with the daily SIC grid. We then calculated sample-level SIC as the mean SIC of all endpoints classified as sea-ice-covered.

We updated Section 2.7 and revised the manuscript to ensure clarity, as follows:

*"In this study, remote sensing data are utilized to illustrate the spatiotemporal distribution of sea ice concentrations (SICs) in the Southern Ocean. For regional-scale visualization of sea-ice extent (SIE) and SIC variability, we used the Sea Ice Index (Version 3) distributed by the National Snow and Ice Data Center*

*(NSIDC) (Fetterer et al., 2017), which is derived from passive-microwave observations from DMSP SSM/I and SSMIS sensors (Cavalieri et al., 1997).*

*Sea-ice concentrations used here are derived from daily gridded passive-microwave SIC products, which provide all-weather coverage and are widely used for polar sea-ice monitoring. The SIC of each sample is calculated using the following formula:*

$$SIC = \frac{\sum_{i=1}^{Ns} SIC_i}{Ns} \quad (7)$$

*where $SIC_i$ represents the average sea ice density at the endpoint of the specified track. Ns represents the total number of trajectory endpoints located on the sea ice area. For each trajectory endpoint, the SIC value was extracted by collocating the endpoint latitude/longitude and the corresponding day with the daily SIC grid; the sample-level SIC_i was then calculated as the mean SIC across all sea-ice-covered endpoints (Ns). Sea ice concentration data are from the AMSR2 dataset (Version 5.4. University of Bremen, Germany. Index of /amsr2/asi_daygrid_swath/s3125).*"

References:

Cavalieri, D. J., Gloersen, P., Parkinson, C. L., Comiso, J. C., and Zwally, H. J.: Observed Hemispheric Asymmetry in Global Sea Ice Changes, Science, 278, 1104–1106, https://doi.org/10.1126/science.278.5340.1104, 1997.

Fetterer, F., Knowles, K., Meier, W. N., Savoie, M. and Windnagel, A. K.: Sea Ice Index. (G02135, Version 3). [Data Set]. Boulder, Colorado USA. National Snow and Ice Data Center. https://doi.org/10.7265/N5K072F8, 2017.

**Comment 2:** L54-55: It may be more appropriate to attribute the elevated ON to sea-ice-associated ecosystems rather than sea ice dynamics.

**Response**: Thank you for raising this important point. We agree that it is more precise to attribute the elevated ON to sea-ice-associated ecosystems rather than broadly to "sea-ice dynamics". We revised the text accordingly.

*"Near Antarctic, ON concentrations and ON/TN ratios were distinctly elevated in sea-ice-influenced air masses, highlighting the role of sea-ice-associated ecosystems as a likely driver of enhanced ON production and emissions."*

**Comment 3:** L98-99: In open oceans, do previous studies suggest a strong correlation between ocean primary productivity and ON or WSON? In polar regions, sea ice variability directly affects primary productivity and consequently ON levels, implying a potential linkage to sea ice dynamics.

**Response**: Previous studies have indeed reported correlations between ocean primary productivity and atmospheric WSON, including in open-ocean environments (e.g., Altieri et al., 2016). We have revised the text to clarify that such relationships have been observed. For polar environments, we now more explicitly describe sea ice variability as a plausible indirect driver of ON by modulating primary productivity

and related emissions.
We revised the main text, as follows:

"*Yet open-ocean and polar regions, where sea ice variability can strongly modulate primary productivity and thus potentially influence ON emissions, remain sparsely observed, limiting constraints on potential sea ice linked controls on ON, especially for high latitudes (Altieri et al., 2016; Matsumoto et al., 2022).*"
Reference:

Altieri, K. E., Fawcett, S. E., Peters, A. J., Sigman, D. M., and Hastings, M. G.: Marine biogenic source of atmospheric organic nitrogen in the subtropical North Atlantic, Proc. Natl. Acad. Sci. U.S.A., 113, 925–930, https://doi.org/10.1073/pnas.1516847113, 2016.

Matsumoto, K., Kobayashi, H., Hara, K., Ishino, S., and Hayashi, M.: Water-soluble organic nitrogen in fine aerosols over the Southern Ocean, Atmos. Environ., 287, 119287, https://doi.org/10.1016/j.atmosenv.2022.119287, 2022.

**Comment 4:** L104: Perhaps both Antarctic and Arctic campaigns should be referenced, as indicated in Section 2.1 on sample collection.
**Response**: Thank you for the suggestion. The text has been revised to explicitly reference both Antarctic and Arctic campaigns, consistent with the description in Section 2.1, as follows:

"*To address these gaps, we measured aerosol ON and IN using samples collected during three Chinese Arctic and Antarctic research expedition campaigns, spanning ~160° of latitude from the Arctic to Antarctica.*"

**Comment 5:** L194: Should the title of Section 2.5 be revised to "Potential Source Contribution Function analysis" for consistency with the main text?
**Response**: Thank you for the suggestion. The section title has been revised to "Potential Source Contribution Function (PSCF) analysis" for consistency with the main text.

**Comment 6:** L222: Why was a 20 km radius selected for the AEC index calculation? Additionally, why were pressures below 850 hPa assigned a Chl-a value of zero? These aspects require clarification.
**Response**: The 20 km radius was selected to balance spatial representativeness with the native resolution of the satellite Chl-a product, ensuring sufficient valid pixels while minimizing spatial smoothing. Chl-a was set to zero when trajectory endpoints were located over land, sea ice, or at pressures below 850 hPa because air masses at these altitudes are generally decoupled from local ocean surface biological activity. We added this content in Section 2.6 of the article as follows:

"*For each trajectory point, Chl-a concentrations (Chla$_i$) were obtained from satellite remote sensing products (Aqua-MODIS, OCI algorithm; 8-day composite, 4 km × 4 km resolution; https://oceancolor.gsfc.nasa.gov/l3/) within a 20-km radius to*

*reduce the influence of missing/cloud-contaminated pixels and pixel-scale noise, while remaining small enough to preserve local marine biological variability relevant to each trajectory point. The 20 km radius approach has been widely adopted in previous studies to mitigate the uncertainty of trajectory endpoints and ensure robust matching with satellite data coverage in previous research (Park et al., 2018; Zhou et al., 2021, 2023). Trajectory endpoints over Antarctica, sea-ice-covered areas, or at pressures < 850 hPa were assigned Chl-a = 0 because air masses at these altitudes are generally decoupled from local ocean surface biological activity (Zhou et al., 2023).*"

Reference:

Park, K., Lee, K., Kim, T., Yoon, Y. J., Jang, E., Jang, S., Lee, B., and Hermansen, O.: Atmospheric DMS in the Arctic Ocean and Its Relation to Phytoplankton Biomass, Global Biogeochemical Cycles, 32, 351–359, https://doi.org/10.1002/2017GB005805, 2018.

Zhou, S., Chen, Y., Paytan, A., Li, H., Wang, F., Zhu, Y., Yang, T., Zhang, Y., and Zhang, R.: Non-Marine Sources Contribute to Aerosol Methanesulfonate Over Coastal Seas, JGR Atmospheres, 126, e2021JD034960, https://doi.org/10.1029/2021JD034960, 2021.

Zhou, S., Chen, Y., Wang, F., Bao, Y., Ding, X., and Xu, Z.: Assessing the Intensity of Marine Biogenic Influence on the Lower Atmosphere: An Insight into the Distribution of Marine Biogenic Aerosols over the Eastern China Seas, Environ. Sci. Technol., 57, 12741–12751, https://doi.org/10.1021/acs.est.3c04382, 2023.

**Comment 7:** L282-284: This raises an important point: could the previously underestimated percentage result from the exclusion of WION? If so, can atmospheric ON fractions, WION and WSON, be quantified or estimated?

**Response:** The previously underestimated ON/TN percentage could partly result from the exclusion of WION. In principle, if both total ON and WSON are measured for the same samples, WION can be calculated by difference (WION = total ON – WSON), and the relative fraction can be expressed accordingly. However, at this stage we are unable to robustly quantify atmospheric ON fractions (WION vs. WSON) for our dataset. First, the very limited aerosol mass in remote/clean environments makes conventional WSON determination (aqueous extraction) difficult and often sample-volume constrained. Second, we are concerned that uncertainties associated with the traditional WSON extraction/measurement approach may propagate into large errors when estimating WION by difference, potentially biasing the inferred fractions. The quantification of WION and WSON partitioning may be an important topic for future work. We appreciate the reviewer's suggestion, which highlights an important methodological issue and will be carefully considered in the future studies.

**Comment 8:** L299-304: Might secondary marine sources involve organic species from biological activities that undergo atmospheric oxidation to form ON? Beyond reactions with acidic species, are there alternative production pathways in the marine boundary layer?

**Response**: Thank you for this insightful suggestion. We agree that the description of secondary marine ON sources require expansion. In the revised manuscript, we have expanded this section to include the oxidative pathways of biogenic volatile organic compounds (BVOCs).

Specifically, we now acknowledge that marine BVOCs (e.g., isoprene and monoterpenes) emitted by phytoplankton undergo atmospheric oxidation via OH radicals (daytime) and $NO_3$ radicals (nighttime) to form secondary ON (Fisher et al., 2016). The $NO_3$-initiated oxidation in the marine boundary layer is a particularly efficient pathway for producing multifunctional organic nitrates (Ng et al., 2017). Furthermore, we have noted that small-molecule alkyl nitrates can also be produced via photochemical reactions of dissolved organic matter (DOM) in the sea surface microlayer and then released into the atmosphere (Chuck et al., 2002).

The revised text is as follows:

"*ON in the MABL primarily originates from two main source pathways: marine emissions and long-distance continental transport. Marine sources include primary ON, predominantly associated with sea-spray particles enriched in biological material from the ocean surface microlayer, and secondary ON. The latter not only derives from marine precursors such as alkylamines that react with acidic species (Altieri et al., 2016; Facchini et al., 2008), but also significantly involves the atmospheric oxidation of marine-derived biogenic volatile organic compounds (BVOCs). Specifically, isoprene and monoterpenes emitted from the ocean can react with hydroxyl (OH) or nitrate radicals ($NO_3$) to form secondary organic nitrates (Fisher et al., 2016; Ng et al., 2017). Additionally, direct sea-to-air emissions of light alkyl nitrates produced photochemically in the surface water contribute to the MABL ON pool (Chuck et al., 2002).*"

References:

Altieri, K. E., Fawcett, S. E., Peters, A. J., Sigman, D. M., and Hastings, M. G.: Marine biogenic source of atmospheric organic nitrogen in the subtropical North Atlantic, Proc. Natl. Acad. Sci. U.S.A., 113, 925–930, https://doi.org/10.1073/pnas.1516847113, 2016.

Chuck, A. L., Turner, S. M., and Liss, P. S.: Direct Evidence for a Marine Source of C1 and C2 Alkyl Nitrates, Science, 297, 1151–1154, https://doi.org/10.1126/science.1073896, 2002.

Facchini, M. C., Decesari, S., Rinaldi, M., Carbone, C., Finessi, E., Mircea, M., Fuzzi, S., Moretti, F., Tagliavini, E., Ceburnis, D., and O'Dowd, C. D.: Important Source of Marine Secondary Organic Aerosol from Biogenic Amines, Environ. Sci. Technol., 42, 9116–9121, https://doi.org/10.1021/es8018385, 2008.

Fisher, J. A., Jacob, D. J., Travis, K. R., Kim, P. S., Marais, E. A., Chan Miller, C., Yu, K., Zhu, L., Yantosca, R. M., Sulprizio, M. P., Mao, J., Wennberg, P. O., Crounse, J. D., Teng, A. P., Nguyen, T. B., St. Clair, J. M., Cohen, R. C., Romer, P., Nault, B. A., Wooldridge, P. J., Jimenez, J. L., Campuzano-Jost, P., Day, D.

A., Hu, W., Shepson, P. B., Xiong, F., Blake, D. R., Goldstein, A. H., Misztal, P. K., Hanisco, T. F., Wolfe, G. M., Ryerson, T. B., Wisthaler, A., and Mikoviny, T.: Organic nitrate chemistry and its implications for nitrogen budgets in an isoprene- and monoterpene-rich atmosphere: constraints from aircraft (SEAC4 RS) and ground-based (SOAS) observations in the Southeast US, Atmos. Chem. Phys., 16, 5969–5991, https://doi.org/10.5194/acp-16-5969-2016, 2016.

Ng, N. L., Brown, S. S., Archibald, A. T., Atlas, E., Cohen, R. C., Crowley, J. N., Day, D. A., Donahue, N. M., Fry, J. L., Fuchs, H., Griffin, R. J., Guzman, M. I., Herrmann, H., Hodzic, A., Iinuma, Y., Jimenez, J. L., Kiendler-Scharr, A., Lee, B. H., Luecken, D. J., Mao, J., McLaren, R., Mutzel, A., Osthoff, H. D., Ouyang, B., Picquet-Varrault, B., Platt, U., Pye, H. O. T., Rudich, Y., Schwantes, R. H., Shiraiwa, M., Stutz, J., Thornton, J. A., Tilgner, A., Williams, B. J., and Zaveri, R. A.: Nitrate radicals and biogenic volatile organic compounds: oxidation, mechanisms, and organic aerosol, Atmos. Chem. Phys., 17, 2103–2162, https://doi.org/10.5194/acp-17-2103-2017, 2017.

**Comment 9:** Regarding "continental sources," does this refer to ON formed over continents and subsequently transported to oceanic regions?

**Response**: In this manuscript, "continental sources" refers to the aggregate influence associated with air masses originating from (or transported over) continental regions. This includes: (1) ON formed directly over land and transported offshore, and (2) ON formed in situ during transport from continental precursors (e.g., anthropogenic $NO_x$ and VOCs). Because our dataset does not allow us to distinguish these two sub-pathways, we use the term "continental sources" to represent their aggregate contribution.

We have revised the text to acknowledge the existence of both mechanisms, as follows:

"*Continental sources involve the long-range transport of organic emissions—including combustion byproducts, soil- and vegetation-derived compounds, and biomass burning aerosols. It is important to note that these continental inputs include both ON formed directly over land and ON produced from continental precursors during transport (Duce et al., 2008; Li et al., 2025). This transport can significantly influence remote ocean regions (Cape et al., 2011; Jickells et al., 2013).*"

References:

Cape, J. N., Cornell, S. E., Jickells, T. D., and Nemitz, E.: Organic nitrogen in the atmosphere — Where does it come from? A review of sources and methods, Atmos. Res., 102, 30–48, https://doi.org/10.1016/j.atmosres.2011.07.009, 2011.

Duce, R. A., LaRoche, J., Altieri, K., Arrigo, K. R., Baker, A. R., Capone, D. G., Cornell, S., Dentener, F., Galloway, J., Ganeshram, R. S., Geider, R. J., Jickells, T., Kuypers, M. M., Langlois, R., Liss, P. S., Liu, S. M., Middelburg, J. J., Moore, C. M., Nickovic, S., Oschlies, A., Pedersen, T., Prospero, J., Schlitzer, R.,

Seitzinger, S., Sorensen, L. L., Uematsu, M., Ulloa, O., Voss, M., Ward, B., and Zamora, L.: Impacts of Atmospheric Anthropogenic Nitrogen on the Open Ocean, Science, 320, 893–897, https://doi.org/10.1126/science.1150369, 2008.

Jickells, T., Baker, A. R., Cape, J. N., Cornell, S. E., and Nemitz, E.: The cycling of organic nitrogen through the atmosphere, Phil. Trans. R. Soc. B, 368, 20130115, https://doi.org/10.1098/rstb.2013.0115, 2013.

Li, Y., Fu, T.-M., Yu, J. Z., Zhang, A., Yu, X., Ye, J., Zhu, L., Shen, H., Wang, C., Yang, X., Tao, S., Chen, Q., Li, Y., Li, L., Che, H., and Heald, C. L.: Nitrogen dominates global atmospheric organic aerosol absorption, Science, 387, 989–995, https://doi.org/10.1126/science.adr4473, 2025.

**Comment 10:** L306: How was it determined that air masses spent 22.6% of their 5-day trajectories over continental areas? Is a single trajectory track used per sample? Methodological details appear insufficient, aligning with general concerns.

**Response**: Thank you for pointing this out. We agree additional methodological detail is needed. The percentage (e.g., 22.6%) represents the time-weighted residence-time fraction of trajectory endpoints over continental grid cells within the 5-day backward trajectories. For the full dataset, we used a representative midpoint for each 48 h sample and generated trajectories at 6 h intervals; residence-time fractions were then computed by classifying endpoints (open ocean/sea ice/continent) and applying an exponential time-weighting to account for dispersion and removal during transport. For the Antarctic sea-ice analysis subset, we additionally applied a moving-track ensemble approach (48 hourly positions per sample) and computed the same residence-time metrics using the ensemble of trajectories.

We revised Section 2.4 to describe this nested strategy and to define the residence-time calculation.

*"To study air mass origins, air mass backward trajectories have been calculated using the Hybrid Single-Particle Lagrangian Integrated Trajectories (HY-SPLIT) model with meteorological fields from the National Oceanic and Atmospheric Administration (NOAA) air resources laboratory GDAS database. Five-day backward trajectories were calculated in order to reveal the history of the air masses arriving at the sampling site (Stein et al., 2015). Each trajectory originated at the vessel's real-time position with an arrival height of 20 m, capturing boundary layer transport while minimizing local ship influence. Air mass backward trajectories were simulated using the HY-SPLIT model with meteorological fields from the NOAA GDAS database to reveal the transport history of air masses arriving at the vessel (Stein et al., 2015). Given that the ship was continuously moving and each sample integrates air masses over approximately 2–4 degrees of latitude, we applied a nested strategy to account for spatiotemporal variability. For the initial characterization of the entire dataset, a representative sampling location was defined for each sample using the average latitude and longitude of its start and end positions, with backward trajectories simulated at 6-h intervals anchored to this midpoint to identify dominant air-mass*

*categories (Fig. 3). Subsequently, to precisely investigate the influence of sea ice on ON in the Southern Ocean and Antarctic marginal regions (Section 4.2), a targeted high-resolution analysis was performed on this subset of samples. For each Antarctic sample, the actual cruise track was equally divided into 48 points corresponding to the hourly intervals of the 48-h sampling period, and a 120-h backward trajectory was calculated for each of these 48 coordinates (Fig. S3a and b).*

*To determine whether the backward trajectories of the MABL samples were mainly influenced by the open ocean, sea-ice-covered regions, or the continental area, we calculated the time-weighted residence-time ratios of air masses over sea ice ($R_S$), open ocean ($R_O$), and the continental area ($R_C$) using the following equation:*

$$R_S(R_O \text{ or } R_C) = \frac{\sum_{i=1}^{N_S(N_O \text{ or } C)} \times e^{-(\frac{t_i}{120})}}{\sum_{i=1}^{N_{total}} \times e^{-(\frac{t_i}{120})}} \qquad (4)$$

*where $N_{total}$ denotes the total number of trajectory endpoints; $N_S$ $N_O$ and $N_C$ represent the numbers of endpoints located over sea ice, the open ocean, and the Antarctic ice sheet, respectively. $t_i$ is the backward-trajectory time (in hours), and $t_i/120$ is a time-weighting factor (Zhou et al., 2021). This factor accounts for air-mass dispersion during transport and aerosol removal by particle deposition; therefore, regions associated with longer trajectory times exert weaker influences on the sampling site, whereas nearby regions exert stronger influences. Accordingly, higher values of $R_S$, $R_O$ and $R_C$ indicate greater influences from sea ice, the open ocean, and the Antarctic ice sheet, respectively.* "

References:

Stein, A. F., Draxler, R. R., Rolph, G. D., Stunder, B. J. B., Cohen, M. D., and Ngan, F.: NOAA's HYSPLIT Atmospheric Transport and Dispersion Modeling System, Bull. Am. Meteorol. Soc., 96, 2059–2077, https://doi.org/10.1175/BAMS-D-14-00110.1, 2015.

Zhou S, Chen Y, Paytan A, et al. Non-Marine Sources Contribute to Aerosol Methanesulfonate Over Coastal Seas [J]. Journal of Geophysical Research: Atmospheres, 2021, 126: e2021JD034960.

**Comment 12:** Figure 4: Note that EC is not an ionic species.
Response: Thank you for noting this. The figure caption and associated text have been corrected.

**Comment 13:** L364-368: Employing a longer time interval in backward trajectory analysis might reveal air masses traversing continents. As demonstrated, EC detected in remote oceans confirms long-range transport from continental sources. Similarly, fine ON particles could be transported to the remote Southern Ocean. The absence of correlation between ON and EC or $nssCa^{2+}$ may indicate that ON is not predominantly of continental origin.
Response: We agree that extending the trajectory duration would provide a more

comprehensive history of air mass origins and is particularly useful for identifying the transport of long-lived species like EC. In this study, we selected a 5-day trajectory interval primarily to align with the typical atmospheric residence time of ON (Textor et al., 2006; Cape et al., 2011). This window allows us to focus on the sources most likely to influence the observed concentrations.

Our chemical tracer data supports the reviewer's insight regarding the distinction between long-range transport and local production. While continental air masses may indeed reach the Southern Ocean (bringing trace EC), the distinct lack of correlation between ON and continental tracers ($nssCa^{2+}$, EC) suggests that the continental contribution to the ON pool is minimal. This decoupling reinforces our conclusion that the observed ON is predominantly driven by local marine processes.

We have revised the text to reflect this understanding, as follows:

"*In the SO region, ON concentrations were the lowest among all regions (mean = 12.0 ± 7.1 ng m$^{-3}$), yet the ON/TN ratio was relatively high (27.8 ± 11.0%). Back-trajectory analysis indicates that air masses predominantly originated from the open ocean and Antarctic continent (Fig. 3d), with minimal anthropogenic influence. ON here exhibited a significant positive correlation with Na$^+$ (Fig. 4; r = 0.43, p < 0.01), but no significant relationships with nssK$^+$, nssCa$^{2+}$ or EC. While long-range transport events may deliver stable continental tracers like EC to this remote region, the lack of correlation between ON and these markers suggests that continental inputs are not the primary driver of ON variability. This pattern, combined with the association with Na$^+$ suggests that primary sea-salt emissions are an important pathway for ON in the SO atmosphere (Matsumoto et al., 2022), likely through the incorporation of marine-derived organic matter into sea-spray aerosols. Meanwhile, the absence of associations with terrestrial tracers further supports the notion that ON in this remote region is controlled primarily by natural marine processes rather than continental or anthropogenic sources (Altieri et al., 2016).*"

References:

Altieri, K. E., Fawcett, S. E., Peters, A. J., Sigman, D. M., and Hastings, M. G.: Marine biogenic source of atmospheric organic nitrogen in the subtropical North Atlantic, Proc. Natl. Acad. Sci. U.S.A., 113, 925–930, https://doi.org/10.1073/pnas.1516847113, 2016.

Cape, J. N., Cornell, S. E., Jickells, T. D., and Nemitz, E.: Organic nitrogen in the atmosphere — Where does it come from? A review of sources and methods, Atmos. Res., 102, 30–48, https://doi.org/10.1016/j.atmosres.2011.07.009, 2011.

Matsumoto, K., Kobayashi, H., Hara, K., Ishino, S., and Hayashi, M.: Water-soluble organic nitrogen in fine aerosols over the Southern Ocean, Atmos. Environ., 287, 119287, https://doi.org/10.1016/j.atmosenv.2022.119287, 2022.

Textor, C., Schulz, M., Guibert, S., Kinne, S., Balkanski, Y., Bauer, S., Berntsen, T., Berglen, T., Boucher, O., Chin, M., Dentener, F., Diehl, T., Easter, R., Feichter, H., Fillmore, D., Ghan, S., Ginoux, P., Gong, S., Grini, A., Hendricks, J., Horowitz, L., Huang, P., Isaksen, I., Iversen, T., Kloster, S., Koch, D., Kirkev,

A., Kristjansson, J. E., Krol, M., Lauer, A., Lamarque, J. F., Liu, X., Montanaro, V., Myhre, G., Penner, J., Pitari, G., Reddy, S., Seland, Ø., Stier, P., Takemura, T., and Tie, X.: Analysis and quantification of the diversities of aerosol life cycles within AeroCom, Atmos. Chem. Phys., 2006.

**Comment 14:** L377-382: Previous studies have reported high primary productivity at sea ice edges.

**Response:** We agree. The sea ice edge is indeed a hotspot for biological activity, characterized by intense phytoplankton blooms and high primary productivity triggered by increased light availability and water column stratification during ice retreat.

In the revised manuscript, we have explicitly incorporated this point:

"*Sea-ice and open-ocean environments create distinct conditions for the production and emission of ON. While sea ice restricts direct air–sea exchange, it hosts specialized microbial communities and accumulates organic matter within brine channels. During melt and ice-edge retreat, this organic material is released into waters characterized by high primary productivity (Arrigo et al., 2008). This biological intensification enriches the surface microlayer and supplies precursors for aerosolization via sea spray and secondary formation (Dall'Osto et al., 2017; DeMott et al., 2016; Galgani et al., 2016; Wilson et al., 2015).*"

References:

Arrigo, K. R., Van Dijken, G. L., and Bushinsky, S.: Primary production in the Southern Ocean, 1997–2006, J. Geophys. Res., 113, 2007JC004551, https://doi.org/10.1029/2007JC004551, 2008.

Dall'Osto, M., Ovadnevaite, J., Paglione, M., Beddows, D. C. S., Ceburnis, D., Cree, C., Cortés, P., Zamanillo, M., Nunes, S. O., Pérez, G. L., Ortega-Retuerta, E., Emelianov, M., Vaqué, D., Marrasé, C., Estrada, M., Sala, M. M., Vidal, M., Fitzsimons, M. F., Beale, R., Airs, R., Rinaldi, M., Decesari, S., Cristina Facchini, M., Harrison, R. M., O'Dowd, C., and Simó, R.: Antarctic sea ice region as a source of biogenic organic nitrogen in aerosols, Sci Rep, 7, 6047, https://doi.org/10.1038/s41598-017-06188-x, 2017.

DeMott, P. J., Hill, T. C. J., McCluskey, C. S., Prather, K. A., Collins, D. B., Sullivan, R. C., Ruppel, M. J., Mason, R. H., Irish, V. E., Lee, T., Hwang, C. Y., Rhee, T. S., Snider, J. R., McMeeking, G. R., Dhaniyala, S., Lewis, E. R., Wentzell, J. J. B., Abbatt, J., Lee, C., Sultana, C. M., Ault, A. P., Axson, J. L., Diaz Martinez, M., Venero, I., Santos-Figueroa, G., Stokes, M. D., Deane, G. B., Mayol-Bracero, O. L., Grassian, V. H., Bertram, T. H., Bertram, A. K., Moffett, B. F., and Franc, G. D.: Sea spray aerosol as a unique source of ice nucleating particles, Proc. Natl. Acad. Sci. U.S.A., 113, 5797–5803, https://doi.org/10.1073/pnas.1514034112, 2016.

Galgani, L., Piontek, J., and Engel, A.: Biopolymers form a gelatinous microlayer at

the air-sea interface when Arctic sea ice melts, Sci Rep, 6, 29465, https://doi.org/10.1038/srep29465, 2016.

Wilson, T. W., Ladino, L. A., Alpert, P. A., Breckels, M. N., Brooks, I. M., Browse, J., Burrows, S. M., Carslaw, K. S., Huffman, J. A., Judd, C., Kilthau, W. P., Mason, R. H., McFiggans, G., Miller, L. A., Nájera, J. J., Polishchuk, E., Rae, S., Schiller, C. L., Si, M., Temprado, J. V., Whale, T. F., Wong, J. P. S., Wurl, O., Yakobi-Hancock, J. D., Abbatt, J. P. D., Aller, J. Y., Bertram, A. K., Knopf, D. A., and Murray, B. J.: A marine biogenic source of atmospheric ice-nucleating particles, Nature, 525, 234–238, https://doi.org/10.1038/nature14986, 2015.

**Comment 15:** L389: "sea-ice–linked" should be corrected to "sea-ice–associated" for consistency.

**Response**: Thank you for the suggestion. The wording has been corrected throughout the manuscript for consistency:

"*Multiple lines of evidence point to sea-ice–associated biological processes as the driver of these enhancements...*"

**Comment 16:** Figure 5 caption: The statement "owing to missing satellite data and the methodologies in Sections 2.6 and 2.7" is ambiguous.

**Response**: Thank you for pointing this out. We clarify that the sample size for the correlations was reduced for two reasons. First, satellite products for SIC and Chl-a are not always available along the trajectories (e.g., due to gaps/clouds). Second, following our data-screening criteria, SIC or Chl-a is considered valid for a sample only when at least 75% of the trajectory points have valid satellite values. We have revised the Figure 5 caption accordingly.

We revised the caption of Figure 5 as follows:

"*Figure 5. Comparison of measured ON concentrations (a) and ON/TN ratio (b) between SI and OO aerosol samples ("**" indicating p < 0.01). And correlations between SIC (c), AEC (d) and ON concentration in SI aerosol samples. The sample sizes are n = 10 for panel (c) and n = 6 for panel (d). These reduced sample sizes are due to unavailable satellite SIC/Chl-a data along the trajectories, and in this study SIC or Chl-a is used only when ≥75% of the trajectory points have valid satellite values (see Sections 2.6–2.7).*"

**Comment 17:** L456-460: If WION was substantially underestimated in prior studies, the current observations hold significant climate relevance, as WION may function as cloud condensation nuclei.

**Response**: We thank the reviewer for highlighting this critical implication. We fully agree that the underestimation of ON in prior studies (due to the omission of WION) has profound consequences for our understanding of aerosol-cloud interactions.

As the reviewer correctly points out, WION components—such as biological colloids and proteinaceous matter—can significantly influence cloud microphysics. They can

act directly as cloud condensation nuclei (CCN) or enhance CCN activity by lowering surface tension (Ovadnevaite et al., 2011). Furthermore, insoluble organic particles are known to be efficient Ice Nucleating Particles (INP) in the marine atmosphere (Wilson et al., 2015).

In the revised manuscript, we have explicitly expanded this section:

*"Comparison with prior WSON-only datasets suggests that earlier studies likely underestimated total ON—by approximately 40% in the Southern Ocean—due to omission of WION. Accounting for both soluble and insoluble phases is therefore essential for constraining nitrogen deposition to the oceans and for representing ON's roles in atmospheric processes. Specifically, given that WION may significantly influence cloud condensation nuclei activity and cloud droplet formation, overlooking this fraction could lead to substantial uncertainties in assessing the radiative forcing and climate effects of marine aerosols."*

References:

Ovadnevaite, J., Ceburnis, D., Martucci, G., Bialek, J., Monahan, C., Rinaldi, M., Facchini, M. C., Berresheim, H., Worsnop, D. R., and O'Dowd, C.: Primary marine organic aerosol: A dichotomy of low hygroscopicity and high CCN activity: MARINE AEROSOL-CLOUD INTERACTIONS, Geophys. Res. Lett., 38, n/a-n/a, https://doi.org/10.1029/2011GL048869, 2011.

Wilson, T. W., Ladino, L. A., Alpert, P. A., Breckels, M. N., Brooks, I. M., Browse, J., Burrows, S. M., Carslaw, K. S., Huffman, J. A., Judd, C., Kilthau, W. P., Mason, R. H., McFiggans, G., Miller, L. A., Nájera, J. J., Polishchuk, E., Rae, S., Schiller, C. L., Si, M., Temprado, J. V., Whale, T. F., Wong, J. P. S., Wurl, O., Yakobi-Hancock, J. D., Abbatt, J. P. D., Aller, J. Y., Bertram, A. K., Knopf, D. A., and Murray, B. J.: A marine biogenic source of atmospheric ice-nucleating particles, Nature, 525, 234–238, https://doi.org/10.1038/nature14986, 2015.

**Comment 18:** Please double-check the Supplementary material. In particular, the full name instead of abbreviation may be better in the Figure captions.
**Response**: Thank you for the reminder. The Supplementary material has been carefully checked, and abbreviations in figure captions have been replaced with full names where appropriate.

**End of responses to the Referee.**